# Doing Experiments and Revising Rules
# with Natural Language and Probabilistic Reasoning

**Wasu Top Piriyakulkij**[1]*  **Cassidy Langenfeld**[1]  **Tuan Anh Le**[2]  **Kevin Ellis**[1]

Cornell University[1]    Google[2]

## Abstract

We give a model of how to infer natural language rules by doing experiments. The model integrates Large Language Models (LLMs) with Monte Carlo algorithms for probabilistic inference, interleaving online belief updates with experiment design under information-theoretic criteria. We conduct a human-model comparison on a Zendo-style task, finding that a critical ingredient for modeling the human data is to assume that humans also consider fuzzy, probabilistic rules, in addition to assuming that humans perform approximately-Bayesian belief updates. We also compare with recent algorithms for using LLMs to generate and revise hypotheses, finding that our online inference method yields higher accuracy at recovering the true underlying rule, and provides better support for designing optimal experiments.

## 1  Introduction

An important way that humans grow their knowledge of the world is by experimentation and other forms of active learning. This process is most clearly present in the experimental sciences, but similar processes of active inference begin in infancy through early childhood [1, 2, 3, 4, 5]. Within everyday adult cognition, active experimentation helps us quickly learn to use new devices and tools.

A basic framework for modeling experimentation is to alternate between conducting a good experiment, and updating one's beliefs based on those experimental results [6]. These beliefs concern a latent *hypothesis* about the regularity or trend the experimenter is investigating. This leaves open at least two computational questions. First, we need to define a hypothesis space. Second, we need efficient algorithms for belief updates and experiment generation. Such algorithms should reason about probabilistic beliefs—considering many hypotheses and their associated probabilities—in order to find experiments that optimally resolve different competing hypothesis.

Here we will introduce a model that represents hypotheses in natural language—even for problems that do not intrinsically involve human language. We do this for two reasons. First, natural language can index many human concepts, and can recursively combine them, giving an expressive hypothesis space. Second, it allows using Large Language Models (LLMs) to aid the inference task of updating beliefs after each experiment, giving tractable, approximate probabilistic inference when we view the LLM as a proposal distribution for a Monte Carlo estimator.

We are especially interested in comparing our model to human behavior, given the long legacy of probabilistic modeling within cognitive science [7, 8]. We find a nuanced picture: vanilla LLMs are not humanlike on our active learning tasks (and underperform humans); our full model outperforms humans; but a simple change—switching from deterministic to probabilistic hypotheses—allows matching humans in overall performance, and agreement with humans on more fine-grained metrics.

From a technical perspective, our work needs to infer natural-language hypotheses in an online setting, so that it can cycle between experimentation and hypothesis formation. This differs from recent

---

*Corresponding author: `wp237@cornell.edu`

38th Conference on Neural Information Processing Systems (NeurIPS 2024).

batched approaches for hypothesis formation [9, 10, 11]. To allow online inference, we hybridize LLMs with Sequential Monte Carlo Samplers (SMC-S: [12]). In SMC-S, one tracks a modest number of hypotheses that serve as (approximate) samples from the posterior. Meanwhile, the LLM focuses the sampler on a small set of candidate hypotheses that it deems relevant, given the data. The resulting sampler facilitates active learning by choosing an experiment which optimally "splits" the candidate hypotheses. With strategies that do not use probabilistic framing, such as tracking a single best-guess hypothesis, the active learner would have little guidance on what experiment to do next.

We will focus here on active inference of basic symbolic concepts expressible in natural language, as we believe these are tractable first targets of study. Concretely, we consider tasks in the spirit of the boardgame 'Zendo', a challenging but accessible game where human players actively learn binary rules combining logical and spatial relations [13, 14, 15], as well as 'Blicket test' style tasks, inspired by studies in developmental psychology [16, 2, 17] that investigate how children learn the causal mechanism behind the activation of a machine. See Figure 1.

We contribute the following:

1. An algorithm for probabilistic inference of latent natural language hypotheses. This derives from SMC-S, but uses an LLM proposal distribution to allow tractable inference over natural language strings, essentially using the LLM to suggest ways of revising the belief state.

2. Model-Human/Model-Baseline comparisons, finding that (1) we get a better fit to human data using natural language, instead of formal languages; (2) the model can be further made more humanlike by considering fuzzy (probabilistic) rules, and (3) that our online inference also yields better accuracy at the actual task relative to recent work [10, 9, 11].

3. Empirical findings about the ability of LLMs to revise hypotheses and propose experiments. On the domains we consider, we find that LLMs are effective for proposing and revising hypotheses, but do not consistently outperform random guessing when proposing experiments.

## 2 Model

We start with standard Bayesian optimal experiment design, which gives a framework for describing both experimentation and hypothesis formation [18, 19]. Our model includes natural-language hypotheses $h \in \Sigma^*$, experiments $x \in \mathcal{X}$, and experiment outcomes $y \in \mathcal{Y}$. We consider equipping $h$ with real-valued parameters $\theta$: For example, if the hypothesized rule is fuzzy (noisy), then $\theta$ would control the noise level. As new experiments are proposed sequentially, we index experiments and outcomes with subscripts, i.e. $x_t$ and $y_t$ for the $t^{\text{th}}$ experiment and outcome, respectively. The objective is to identify ground-truth $h^*$, and to accurately predict the outcome of future experiments.

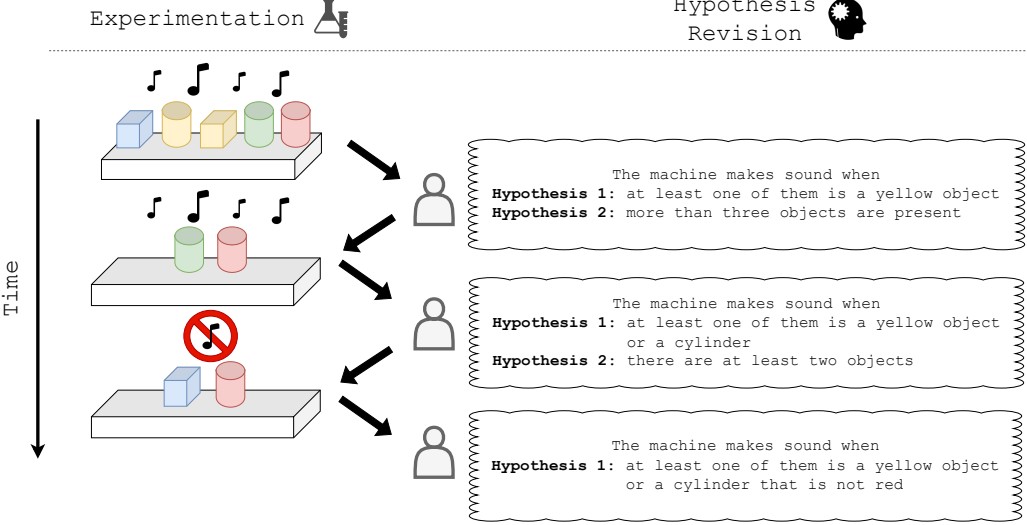

Figure 1: Alternation of experimentation and hypothesis generation on a simplified version of our ActiveACRE domain. Hypotheses characterizes what causes the machine to activate (make noise).

The joint distribution over hypothesis $h, \theta$ and outcomes $y_{1:T}$, given experiments $x_{1:T}$, is

$$p(h, y_{1:T}, \theta | x_{1:T}) = p(h)p(\theta) \prod_{1 \leq t \leq T} p(y_t | x_t, h, \theta) \tag{1}$$

where the prior $p(h)$ favors shorter or simpler hypotheses. From eq. (1) the posterior is

$$p(h | x_{1:T}, y_{1:T}) \propto p(h) \int_\theta p(\theta) \prod_{1 \leq t \leq T} p(y_t | x_t, h, \theta) \mathrm{d}\theta \tag{2}$$

where we assume the above integral is tractable, because $\theta$ is low-dimensional. Ultimately, the purpose of the hypothesis is to make predictions on new experiments. Given a test experiment $x_\text{test}$, an ideal learner predicts an outcome $y_\text{test}$ distributed as follows:

$$p(y_\text{test} | x_\text{test}, x_{1:T}, y_{1:T}) = \sum_h p(h | x_{1:T}, y_{1:T}) \int_\theta p(\theta | h, x_{1:T}, y_{1:T}) p(y_\text{test} | x_\text{test}, h, \theta) \mathrm{d}\theta \tag{3}$$

The optimal experiment for identifying $h$ maximizes the following information gain [20]:

$$x^* = \arg\max_{x \in \mathcal{X}} \mathbb{E}_{p(y | x_{1:T}, y_{1:T}, x)} [D_\text{KL}(p(h | x_{1:T}, y_{1:T}, x, y) || p(h | x_{1:T}, y_{1:T}))] \tag{4}$$

The above computations are intractable because they involve considering the infinitely large set of all hypotheses and experiments. We next describe our LLM-guided approximation methods.

## 2.1 Revising Rules: Online Inference

We introduce a generalization of the Sequential Monte Carlo Sampler (SMC-S) [12], an online approximate inference algorithm which tracks a small pool of hypotheses—called particles—that evolve over time as new data is collected. Tracking representative high-posterior particles allows approximate inference (eq. (2)) and prediction (eq. (3)) by only considering the current particles. This makes the model "boundedly rational" [21]: as the bound on computation (# particles) grows large, the sampler better approximates optimal inference. To the extent that our work offers a cognitive model, we are claiming that humans only consider a small number of hypotheses, which evolve in ways that approximate probabilistic reasoning. This should be seen within the tradition of using approximate inference methods to give mechanistic accounts of human learning [22, 23, 24, 25].

Standard SMC-S tracks $n$ particles at each time point $t$, written $H_t = \{h_t^{(i)}\}_{i=1}^n$. Each particle has a weight, $W_t = \{w_t^{(i)}\}_{i=1}^n$, giving the approximate posterior $p(h | x_{1:t}, y_{1:t}) \approx \sum_i w_t^{(i)} \mathbb{1}\left[h = h_t^{(i)}\right]$. Upon observing a new data point, the particles $H_t$ are pushed through a forward kernel $q_{t+1}(h_{t+1} | h_t^{(i)})$, which randomly perturbs the particles, to obtain new particles $H_{t+1}$. Next, the particles are reweighed to obtain $W_{t+1}$. Finally, a resampling step can be executed to prune low-weight particles and multiply high-weight particles.

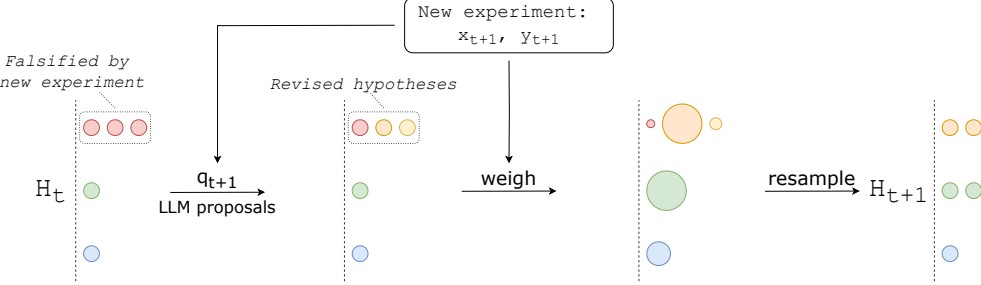

Figure 2: Sequential Monte Carlo method tracks a small number of hypotheses (called particles), each of which is a natural language rule, represented above by circles. After each experiment, the particles are revised in light of the new data by pushing the particles through the forward kernel. Then, the new particles are reweighed according to how well each explains the data we have seen so far. Resampling prunes low-probability hypotheses while multiplying high-probability ones.

Here $h$ is a natural language string, suggesting an LLM should define $q_t(h_t|h_{t-1}^{(i)})$. For example, an LLM can be prompted with a hypothesis, together with the latest experiment outcome, and asked to revise that hypothesis. But calling an LLM to perturb every single particle is expensive, and unnecessary for hypotheses that already explain the data well.

We therefore design a variant of SMC-S whose forward kernel looks globally at the current set of particles and prompts an LLM to revise the worst (lowest-likelihood) particles, while keeping unchanged the best (highest-likelihood) particles. This concentrates the computation on improving bad hypotheses, instead of wasting effort altering what already works. Within the context of LLMs, this can be seen as an online, probabilistic version of hypothesis refinement [9, 26, 10]. Within the context of SMC-S, this mathematically corresponds to defining a forward kernel that conditions on the entire set of previous particles and all seen data points, $q_t(h_t|H_{t-1}, x_{1:t}, y_{1:t})$.[2] Below we formalize our new SMC-S variant, which we call LLM-SMC-S, illustrated in Figure 2.

**Procedure: LLM-SMC-S (A.4).** Given $H_t, W_t$ where $p(h|x_{1:t}, y_{1:t}) \approx \sum_i w_t^{(i)} \mathbb{1}\left[h = h_t^{(i)}\right]$:

1. Define unnormalized target densities $\gamma(h) = p(h, y_{1:t}, x_{1:t})$ and $\gamma'(h) = p(h, y_{1:t+1}, x_{1:t+1})$.
2. Sample $h' \sim q_{t+1}(\cdot|H_t, x_{1:t+1}, y_{1:t+1})$ (i.e., using LLM to revise hypotheses)
3. Compute the weight $w'$ for $h'$ following

$$w' = \frac{A(h', H_t, W_t)}{q_{t+1}(h'|H_t, x_{1:t+1}, y_{1:t+1})} \text{ where } A(h', H_t, W_t) = \frac{1}{n}\sum_{i=1}^{n} w_t^{(i)} \frac{\gamma'(h')r(h_t^{(i)}|h')}{\gamma(h_t^{(i)})} \quad (5)$$

   with the reverse kernel $r(h|h')$ defined as uniform up to strings of a maximum length.
4. Repeat steps 2-3 (sampling/weighing) a total of $n$ times, and normalize the weights. Optionally, resample to generate an unweighted posterior (we always resample).
5. Output: $H_{t+1}$ and $W_{t+1}$, formed from $n$ samples of $h', w'$ with $w'$ normalized from step 4, which approximate $p(h|x_{1:t+1}, y_{1:t+1})$.

The correctness of the above procedure is most easily understood using the following definition:

**Definition: Proper Weighting** [27]**.** Let $\gamma(h)$ be an unnormalized target density, which we can evaluate. Let the corresponding normalized target density be $\pi(h) = \frac{\gamma(h)}{Z_\pi}$ where $Z_\pi = \int \gamma(h)\mathrm{d}h$ is the normalization constant. A weighted particle $h, w$ is properly weighted with respect to $\gamma$ if for any function $f$,

$$E[wf(h)] = Z_\pi E_{\pi(h)}[f(h)]$$

**Proposition 1.** If $H, W$ input to Procedure LLM-SMC-S is properly weighted with respect to $\gamma$, then the output $h', w'$ is properly weighted with respect to $\gamma'$. (Proof in Appendix A.1.)

## 2.2 Doing Experiments: Active Learning

Our active learning works by doing an experiment that maximizes information gain (eq. (4)). Experiments may be complex, such as involving putting objects or instruments in specific positions, and there might be combinatorially many possible experiments. For a rich space of experiments, a bounded learner—human or AI—cannot consider all possibilities.

We will propose experiments using an LLM, but then reassess those proposals under probabilistic criteria. Particularly, we provide an LLM with the hypotheses tracked by the SMC-S sampler at each iteration, and prompt it to generate experiments that support and falsify each hypothesis. Empirically, this process yields a diverse pool of experiments. We take the best experiment proposed by the LLM, as measured under the approximate posterior from SMC-S:

$$x_{t+1} = \underset{x \in \text{PROMPT}(H_t)}{\arg\max} \; \underset{\hat{p}(y|x,x_{1:t},y_{1:t})}{\mathbb{E}} \left[D_{\text{KL}}(\hat{p}(h|x_{1:t}, y_{1:t}, x, y)||\hat{p}(h|x_{1:t}, y_{1:t}))\right] \quad (6)$$

where $\hat{p}$ is approximated with the weighted particles from SMC-S.[3]

---

[2]We note that whether we condition on the seen data points or not does not change the proof. The main novelty of this new variant lies in how $q$ can be conditioned on the entire set of particles $H_{t-1}$

[3]We let the particles $H_t$ be the support for both distributions so that we can calculate KL divergence.

## 2.3 Instantiating the model

All of our experiments have binary outcomes ($y \in \{0, 1\}$), and all of our natural language hypotheses correspond to rules that predict whether an experiment succeeds or fails (1 or 0). Although the rules predict hard all-or-none judgments, a learner can relax that constraint by assuming that the underlying rule is fuzzy (noisy). Many natural language facts and rules actually only partly hold, such as *birds fly* (almost always true), or *birds lay eggs* (true half the time). To handle the possibility of fuzzy rules, we equipped each hypothesized rule with real-valued parameters $\theta$ that control the noise level. The noise parameters decompose into a pair $\theta = (\epsilon, \delta)$ controlling the rate of false-positives/false-negatives:

$$p(y = 1 | x, h, \epsilon, \delta) = \left[ \begin{array}{ll} \delta & \text{if } h(x) = 1 \\ 1 - \epsilon & \text{if } h(x) = 0 \end{array} \right]$$

Under this formulation, hard rules corresponds to $p(\epsilon)$ and $p(\delta)$ having non-zero probability only at value 1. For probabilistic, fuzzy rules, we use Gaussian priors for $p(\epsilon)$ and $p(\delta)$, truncated to [0.5,1], and with a bias toward larger $\epsilon$. The prior $p(h)$ is defined as inversely proportional to wordcount, giving a gentle bias toward parsimony. We investigate both hard and fuzzy rules in our experiments.

Evaluating $h(x)$ requires checking the natural language string $h$ against experiment $x$, for which we use GPT-3.5 to translate the natural language $h$ to code which is run on $x$. We use GPT-4 Turbo to propose hypotheses [30]. Recent studies find a similar breakdown of LLMs works well [9, 10, 11].

## 3 Experimental Results

**Domains.** **Zendo** is a game where a player seeks to infer a hidden binary rule about scenes of colored shapes. Our Zendo games begin with showing the player a positive example scene, followed by 7 rounds of experimentation, where the player builds a scene, and receives feedback on if the scene obeys the hidden rule. After the experimentation phase, players are tested on 8 test scenes, half of which follow the hidden rule. Our setup follows Bramley et al.[13], but modified for LLMs by presenting scenes as text describing each block by its color, size, orientation, groundedness, and what other blocks it touches and stacks (Figure 3).

Our second domain, **ActiveACRE**, derives from The Abstract Causal REasoning (ACRE) dataset [17], which in turn derives from 'blicket' tests in developmental cognitive psychology [16]. The original ACRE is a causal induction dataset where each task is to figure out what causes the 'blicket' machine to make sounds when multiple objects are put on the machine. We add active learning to ACRE: rather than passively observe examples, our ActiveACRE allows the player to try 7 experiments, after passively witnessing the outcome of one experiment involving eight objects. The player is then tested (without further feedback) on all possible combinations of the original eight objects.

**Model-Baseline comparisons.** Table 1 contrasts the performance of different models, showing that online inference with hard rules outperforms all other models on both datasets, including a ReAct-style baseline [31] (Direct LLM), and batched inference with refinement, an approach advocated for in recent work [10, 9]. To measure accuracy on Zendo, we compute the predictive posterior accuracy summed over the 8 test scenes and averaged over all tasks. Because the test set on ActiveACRE

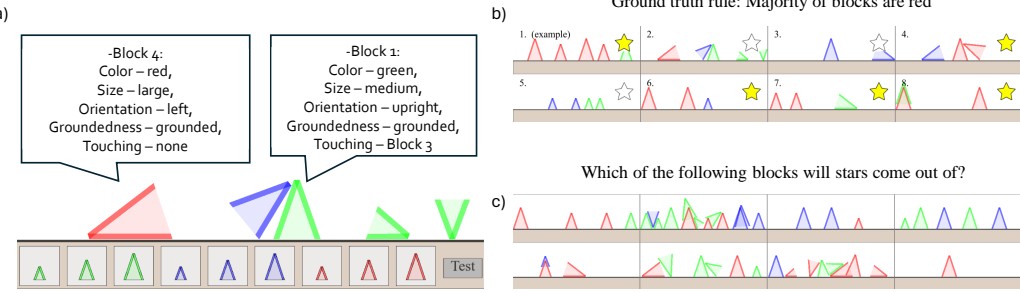

Figure 3: (a) Example Zendo scene and its serialization into text. (b) Eight experiments, each of which is a scene, with a binary outcome (whether the scene makes stars come out of it). (c) Test scenes that evaluate whether a model or human has correctly inferred the hidden rule.

| Method | Zendo | ActiveACRE | | | |
| | Avg Pred Posterior | Avg Pred Posterior | ROC AUC | F1 | Task Solving |
|---|---|---|---|---|---|
| Human from [13] | 5.26 | - | - | - | - |
| Direct LLM [31] | $4.60 \pm 0.19$ | $0.83 \pm 0.05$ | $0.60 \pm 0.02$ | $0.86 \pm 0.04$ | $0.00 \pm 0.00$ |
| Batch, Fuzzy | $4.57 \pm 0.15$ | $0.64 \pm 0.02$ | $0.84 \pm 0.02$ | $0.84 \pm 0.04$ | $0.00 \pm 0.00$ |
| Online, Fuzzy (Ours) | $5.35 \pm 0.09$ | $0.72 \pm 0.01$ | $\mathbf{0.90 \pm 0.03}$ | $0.96 \pm 0.01$ | $0.15 \pm 0.08$ |
| Batch, Hard | $6.01 \pm 0.19$ | $\mathbf{0.89 \pm 0.03}$ | $0.77 \pm 0.04$ | $0.96 \pm 0.01$ | $0.10 \pm 0.07$ |
| Batch w/ Refinement, Hard [9, 10] | $6.18 \pm 0.14$ | $0.86 \pm 0.04$ | $0.73 \pm 0.04$ | $0.91 \pm 0.04$ | $0.15 \pm 0.08$ |
| Online, Hard (Ours) | $\mathbf{6.55 \pm 0.13}$ | $\mathbf{0.92 \pm 0.03}$ | $0.87 \pm 0.04$ | $\mathbf{0.98 \pm 0.01}$ | $\mathbf{0.35 \pm 0.11}$ |

Table 1: Performance on Zendo and ActiveACRE. The results for Zendo are mean $\pm$ standard error of predictive posterior accuracy summed over the test scenes, averaged over the tasks and 5 seeds. ActiveACRE results are mean $\pm$ standard error of each metric averaged over 20 tasks. ActiveACRE is heavily class-inbalanced, so we compute a wider variety of accuracy metrics.

are highly imbalanced, we also report ROC AUC, F1, and task solving scores. The last metric, task solving, measures whether the models perfectly solves each task. The results, especially the large gap on average task solving between our online inference algorithm and batch inference, demonstrate that our online algorithm is more successful at inducing the correct causal law within ACRE, and more accurate at predicting what scenes obey the rule in Zendo. Interestingly, our most performant models—which assume hard deterministic rules—actually surpass human accuracy [13]. This raises the question of how humanlike the model is (or isn't), which we investigate next.

**Model-Human comparisons.** We run the model on the same Zendo games that human participants did, taking human data from Bramley et al. [13]. Average human accuracy is 5.26/8, which surpasses a ReAct-style agent (4.60/8), falls short of our strongest model (6.55/8), and is close to the variant of our model which uses probabilistic fuzzy rules (5.35/8). For a more fine-grained understanding of how human and model accuracy compare, we split accuracy across each of the 10 rules on test scenes that either obey the hidden rule (Rule Following or *RF* condition) or violate the rule (Not Rule Following or *Not RF* condition) (Figure 4a). With fuzzy rules, the

| Method | LogL |
|---|---|
| [13]'s best model | $-1539$ |
| Batch, Fuzzy | $-1660.90$ |
| Online, Fuzzy | $\mathbf{-1478.82}$ |
| Batch, Hard | $-2921.00$ |
| Batch w/ Refinement, Hard | $-3499.76$ |
| Online, Hard | $-5209.93$ |

Table 2: Log likelihood of human data on models summed over all test scenes of all Zendo rules.

model explains 57% of the variation in this more fine-grained measurement of human accuracy ($R^2 = .57$). Switching to hard rules drops this to $R^2 = .10$, suggesting that hard all-or-none rules do not provide as good of an explanation of human behavior, even though hard rules outperform probabilistic ones in terms of accuracy. Doing batch inference instead of online inference degrades fit to $R^2 = .05$. Having the LLM play Zendo directly (ReAct [31]) is only loosely correlated with human accuracy patterns ($R^2 = .25$). We last consider predicting every single human judgment on every single test scene, for every single rule. The online, fuzzy rules model predicts these human judgments at the level of $R^2 = .35$, and importantly, it is only with combination of online inference with fuzzy rules that gives a significant fraction of explained variance (Figure 5), and which assigns the highest likelihood to the raw human data (Table 2).

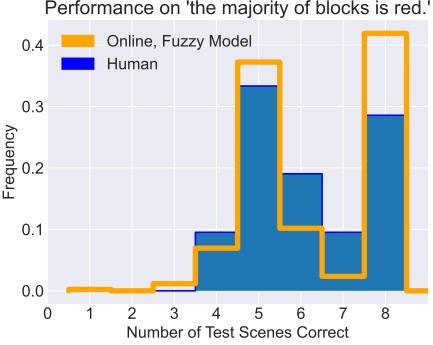

Figure 6: Performance of human and model on 'the majority of blocks is red'

We noticed across many rules a significant difference in human accuracy on RF and Not RF test scenes. Whether people finds RF or Not RF test scenes easier depends on the underlying rule. Figure 4b illustrates this phenomenon and compares it against what each model thinks should be the easier condition. Online learning of fuzzy rules successfully predicts the direction of almost all of these trends, unlike the alternative models.

Hence, we hypothesize that although the hidden Zendo rules are deterministic, humans might nonetheless infer fuzzy rules. Real-world regularities are seldomly deterministic, so it may be rational for human learners to seek probabilistic explanations, especially when they are uncertain about the underlying rule. However, fuzzy

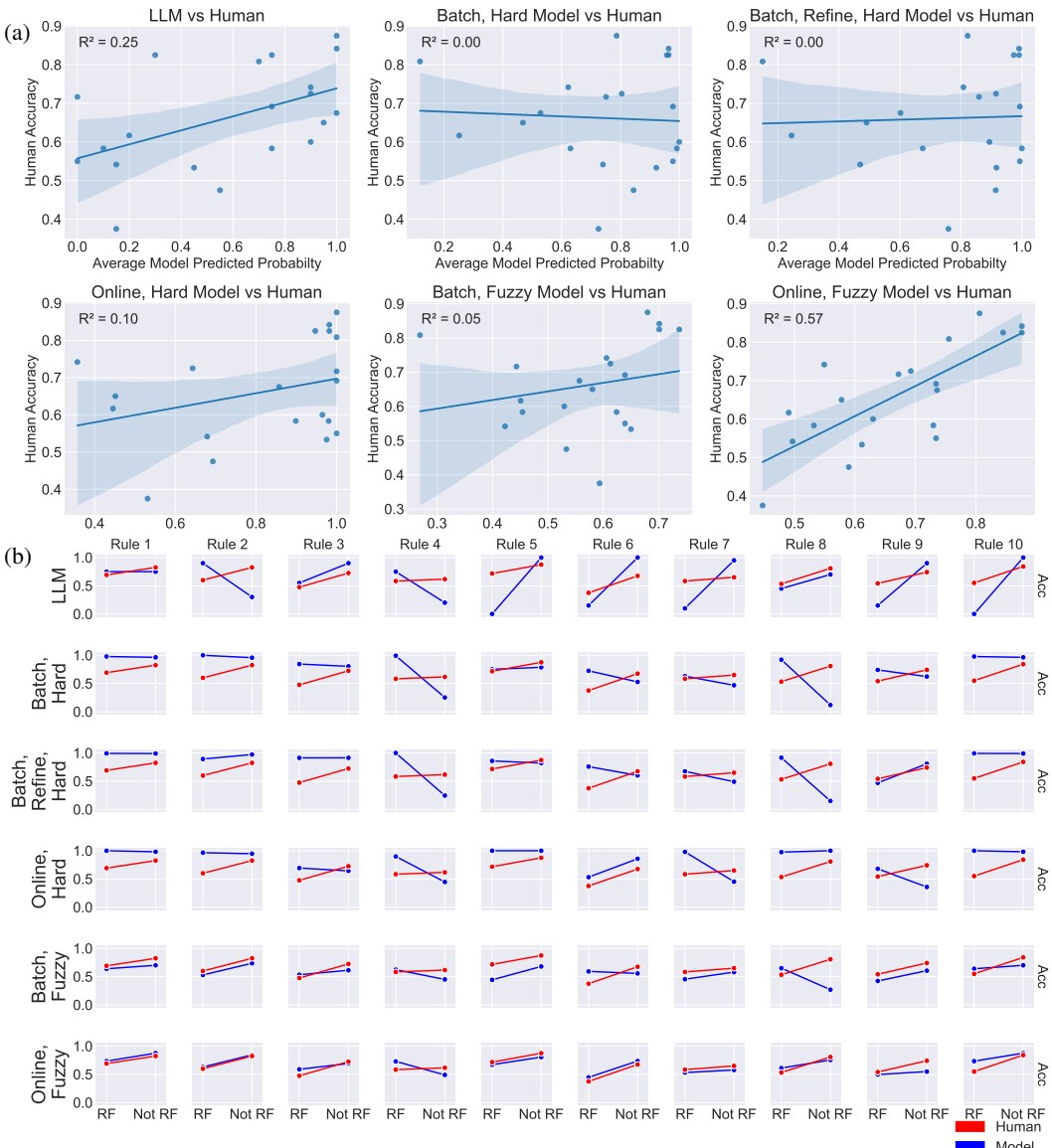

Figure 4: Human vs model accuracy binned by 4 rule-following (RF) and 4 not rule-following (Not RF) test scenes. (a) Each point is a RF or Not RF accuracy for the 10 rules. (b) Rows/columns are methods/rules. Online inference with fuzzy rules (last row) most closely matches humans.

rules on their own do not suffice to explain human judgments: Only by combining with online probabilistic inference do we begin to explain the data.

**Why reason in natural language instead of a formal language?** Many Bayesian models account for human concept learning using probabilistic reasoning over formal languages such as logic [32, 33, 34, 35, 36]. Instead, our model operates over natural language. This helps address two liabilities of formal representations: expressivity and tractability. A handcrafted formal language is often insufficiently expressive, accidentally excluding many human concepts. This expressivity must be limited because, although there exist highly expressive formal languages, in practice, inference in such languages is generally intractable—a tradeoff partly addressed by using LLM proposal distributions.

To illustrate these points, we study a new Zendo rule—'the majority of blocks is red'—which is not expressible in the formal language introduced by [13]. We collect new human data in an IRB-approved study. Figure 6 shows that both humans and our model correctly learn this rule $30\% - 40\%$ of the time. This indicates both the model and humans are able to represent this rule in their hypothesis space, which is unrepresentable in a formal language designed specifically for Zendo.

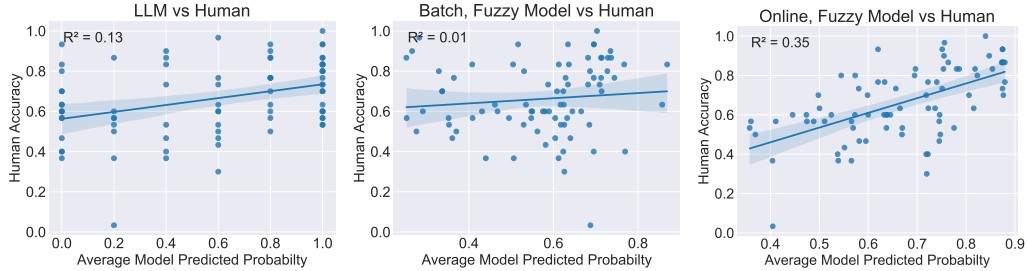

Figure 5: Comparing human and model prediction on each test scene after 7 rounds of experimentation; see also Table 2. Each point is a prediction on a test scene. We only present LLM, best batch model, and best online model here. Please see the figure for all methods at Figure 14.

| Active learning method | Inference method | |
|---|---|---|
| | Online, Fuzzy | Online, Hard |
| LLM | $4.52 \pm 0.08$ | $4.72 \pm 0.08$ |
| Random | $5.03 \pm 0.11$ | $5.84 \pm 0.19$ |
| InfoGain | $\mathbf{5.35 \pm 0.09}$ | $\mathbf{6.55 \pm 0.13}$ |

| Proposer for InfoGain | Number of candidate experiments | | |
|---|---|---|---|
| | 1 | 5 | 10 |
| Random proposer | $5.84 \pm 0.19$ | $6.23 \pm 0.16$ | $6.55 \pm 0.28$ |
| LLM proposer | $5.73 \pm 0.16$ | $6.19 \pm 0.12$ | $6.55 \pm 0.13$ |

Table 3: Average predictive posterior (standard error computed over 5 seeds) of online inference models with different active learning methods on Zendo.

Table 4: Average predictive posterior (standard error computed over 5 seeds) of online inference with hard rules model with different experiment proposers on different number of candidate experiments on Zendo.

Another reason to use natural language representations is that LLMs, trained on human-generated data, may to some extent capture human bias, judgement, and opinions [37, 38, 39]. Unlike approaches based on estimating probabilities on formal languages, incorporating LLMs into our models might therefore make them display more human-like behaviors—as shown in earlier sections—without access to additional human data. Indeed, Table 2 shows that our best-performing model surpasses [13]'s model on human data log likelihood even though the latter fits their models on both human active queries and predictions, while our model does not perform such parameter fitting.

**Bounded rationality.** To understand the effect of computational cost on the results, we analyze performance and human-model fit while varying the computational budget, as measured by LLM calls. Figure 7 plots human-model fit as compute budget varies (see also Table 5). We observe an (inverted) U-shaped curve: Too little budget gives a bad fit, but overshooting also degrades fit. This result aligns with the theory of bounded rationality [21], which argues for considering human's limited cognitive resources, and with the rational analysis of human processing limitations [23, 40].

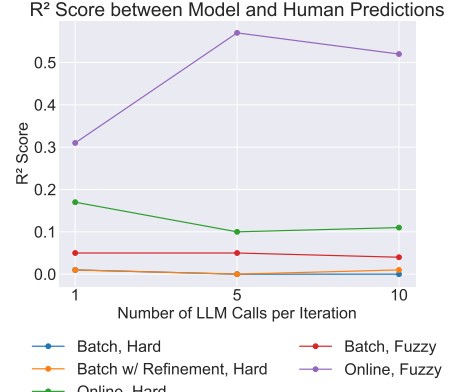

Figure 7: $R^2$ score of human vs model accuracy at different computational budgets. A LLM call batch-samples 15 hypotheses.

**What makes good experiments: LLMs, or Information Gain?** We first study the importance of the information gain objective (Table 3), contrasting three different active learning methods: *LLM* (prompting with the hypotheses and asking for a good experiment); *Random* (handcoded random generator), and *InfoGain* (main method, with LLM proposing experiments). Substituting InfoGain with alternative methods significantly degrades model performance. Reranking LLM proposals with information gain is important, and an LLM—on its own—does not generate experiments that are as effective.

Is this explained by the strength of the LLM experiment proposer, or by the strength of the InfoGain objective? While earlier results support LLMs' effectiveness as hypothesis proposers, Table 4 demonstrates that a random proposer, hand-designed under reasonable assumptions, performs similarly to an LLM experiment proposer. This finding is in line with [41] which argues that LLMs may not always produce the most useful set of candidate questions.

# 4 Related Work

**Bayesian Concept Learning in Cognitive Science.** Bayesian models of few-shot learning of concepts and categories has a long legacy [32, 34, 42, 43, 44, 13, 14, 15, 33, 35, 45, 46, 47, 36]. On Zendo, [13, 14, 15] also engineers a probabilistic context-free grammar to define the Bayesian model to explain Zendo human data; inference is intractable, and they study various approximate inference algorithms on the task. Our work opts for natural language as the hypothesis space for its expressiveness and leverages LLMs to help with approximate inference.

**Inductive Reasoning with Large Language Models.** There are a number of recent works on inductive reasoning with LLMs [10, 9, 56, 11, 57]. [10, 9] study the inductive reasoning ability of vanilla LLMs and propose a simple refinement algorithm to improve performance. [11] explicitly treats LLMs as importance samplers and shows that their model, with prior learned from human data and importance sampling as their inference algorithm, is human-like on some domains. Our work frames many of these works as batch inference—in the spirit of importance sampling, from a probabilistic view—which is compared in our results. Additionally, we model the full life-cycle of experimentation and hypothesis revision.

**Active Learning with Large Language Models.** The problem of performing active learning with the help of LLMs has been studied under the task of asking better questions with LLMs [58, 59, 60, 41, 61]. GATE [58] directly prompts LLMs to ask open-ended, informative questions. Other works [59, 60, 41] use LLMs to help propose several candidate questions, and then use expected information gain to select the question to be asked. Following these previous works, we investigate the relevance of classic criteria from active learning such as expected information gain.

**Probabilistic Inference with Large Language Models.** The framework of probabilistic inference has been applied to LLM-based algorithms. Language-model cascades [62] provides a unifying framework for seeing recent inference-time LLM algorithms [63, 64, 65] as reasoning with probabilistic programs. Other work [66] fine-tunes CoT models by formulating the problem as maximum marginal likelihood where the marginalization over the latent chain of thought is done via probabilistic inference. We use an LLM as an aid for SMC-S, but others have explored using SMC as an aid for LLM decoding [67, 68], which although technically very different, is conceptually complementary.

# 5 Limitations and Next Steps

The work presented is limited in important ways that suggest next steps. Most immediately, many of the the hypotheses we consider are simple and stereotyped in form, and much of the promise of using natural language is that it exposes a rich, expressive representation for combining and creating new ideas [70]. More ambitiously, hypothesis generation, both in science and in everyday thinking, often involves conjecturing the existence of unseen objects, not just unknown regularities, and incorporating this abductive thinking—which is absent from our model—could open many directions.

Although the work here is most directly an account of human behavior within the context of the game Zendo, our model is also more broadly inspired by the mental activities of experimental scientists as they build theories and models, weigh hypotheses, and design experiments. Two basic features of our approach reflect scientific experimentation and theory building. First, scientific theories apply only within a particular regime. For example, many equations in physics only apply when objects are moving slowly, and many thermodynamic equations only apply at equilibrium. Outside this regime, the theory is no longer predictive. Similarly, a variant of a model like ours could hypothesize fuzzy, probabilistic rules which either predict a category with high confidence, or outside the regime in which the rule applies, can fail to make a decisive prediction.

The second way in which this work reflects the practices of experimental science is that it builds its hypotheses via an incremental evolutionary process. In this way, our model is best thought of as performing what is sometimes called 'normal science' — where one works within an existing paradigm, and considers piecemeal evidence — and does not model paradigm shifts [71] (what a scientific theorist might pursue) or deeper conceptual changes [3] (what happens during child development), both of which require deep reanalysis of a broad batch of past data, rather than online incremental revision.

# 6   Acknowledgements

We are grateful to Neil Bramley and Jan-Philipp Fränken for providing the Zendo data, helpful discussions, and comments on the manuscript. We also thank Hao Tang, Simon Alford, Celine Lee, and the anonymous reviewers for valuable comments on the manuscript. This work was supported by an NSF CAREER grant.

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

# A Appendix

## Contents

## A.1 Proof for the weight update of LLM-SMC-S

**Proposition 1.** If $H, W$ input to Procedure LLM-SMC-S is properly weighted with respect to $\gamma$, then the output $h', w'$ is properly weighted with respect to $\gamma'$.

*Proof.* Let $Z_{\pi'} = \int \gamma'(h')dh'$ be the normalizing constant of $\gamma'$ and $\pi'(h') = \frac{\gamma'(h')}{Z_{\pi'}}$ be the normalized target. We want to show

$$E[w'f(h')] = Z_{\pi'}E_{\pi'(h')}[f(h')].$$

Following arguments similar to [29], we have

$$\text{LHS} \tag{7}$$

$$= E_{H_t, W_t, h', x_{1:t+1}, y_{1:t+1}} \left[ \frac{A(h', H_t, W_t)}{q_{t+1}(h'|H_t, x_{1:t+1}, y_{1:t+1})} f(h') \right] \qquad \text{(sub in the definition of } w) \tag{8}$$

$$= E_{H_t, W_t} \left[ \int A(h', H_t, W_t)f(h')dh' \right] \qquad \text{(write } E_{h' \sim q_{t+1}} \text{ as an integral)} \tag{9}$$

$$= E_{H_t, W_t} \left[ \frac{1}{n} \sum_{i=1}^{n} w^{(i)} \frac{\gamma'(h')r(h^{(i)}|h')}{\gamma(h^{(i)})} f(h')dh' \right] \qquad \text{(sub in the definition of } A) \tag{10}$$

$$= \frac{1}{n} \sum_{i=1}^{n} E_{h^{(i)}, w^{(i)}} \left[ \int w^{(i)} \frac{\gamma'(h')r(h^{(i)}|h')}{\gamma(h^{(i)})} f(h')dh' \right] \qquad \text{(pull the } \sum \text{ out of the } \int) \tag{11}$$

$$= \frac{1}{n} \sum_{i=1}^{n} E_{h^{(i)}, w^{(i)}} [w^{(i)}g(h^{(i)})] \qquad \text{(denote the integral as } g(h^{(i)})) \tag{12}$$

$$= Z_{\pi}E_{\pi(h)}[g(h)] \qquad \begin{array}{l}\text{(apply the proper weighting} \\ \text{property with test function } g)\end{array} \tag{13}$$

$$= Z_{\pi} \int \pi(h) \int \frac{\gamma'(h')r(h|h')}{\gamma(h)} f(h')dh'dh \qquad \text{(sub in expression for } g) \tag{14}$$

$$= \int \int \gamma'(h')r(h|h')f(h')dh'dh \qquad \text{(cancel terms using } Z_{\pi} = \gamma/\pi) \tag{15}$$

$$= \int \gamma'(h')f(h')dh' \qquad (r(h'|h) \text{ is normalized}) \tag{16}$$

$$= Z_{\pi'}E_{\pi'(h')}[f(h')] \qquad (Z_{\pi'} = \gamma'/\pi') \tag{17}$$

$$= \text{RHS}. \tag{18}$$

## A.2 Code and data availability

Code and data available at

`https://github.com/topwasu/doing-experiments-and-revising-rules/`

## A.3 Zendo and ACRE details

**Zendo.** Zendo is a game where a player seeks to infer a hidden binary rule about assemblies of colored blocks. The game starts by providing the player with a positive scene that follows the hidden

rule. Then, the player queries an oracle as to a particular scene follows the rule or not, or makes a guess about the secret rule. The game ends when they guess correctly.

Bramley et al. (2018) [13] introduces a 2D version of the Zendo game shown in Figure 3. The scenes consist of blocks, each with its own color (red, blue, green) and size (small, medium, large). The blocks can have different orientations and positions in a 2D scene. They may and may not touch each other. The game starts with an initial phase where a rule-following scene is given, followed by 7 rounds of active learning phase where the player gets to query an oracle for ground truth prediction. At the end, the player enters a prediction phase where they are asked to give predictions for 8 test scenes (4 rule-following and 4 not rule-following). Bramley et al. (2018) study human gameplay on 10 rules, collecting data from 30 participants who each play Zendo 10 times (once per rule). They use a cover story of an alien planet where some arrangements of blocks emit radiation, and the task is to figure out a rule predicting radiation emission.

Zendo is most naturally framed as a visual-physical concept learning problem. For our model, however, we will work with discrete symbolic descriptions of scenes. This makes that problem more compatible with the language-of-thought paradigm, and also allows using LLMs to operationalize the language of thought. We therefore modify Bramley et al. (2018)'s version of Zendo by associating each block in a scene with discrete attributes instead. The five attributes are color (red, blue, green), size (small, medium, large), orientation (upright, left, right, strange), groundedness (grounded, ungrounded, stacking), and touching (which blocks it touches / stacks). While this natural language version of the game removes continuous attributes, such as $x, y$ position and orientation in 2D space, from its scene representations, these five attributes still maintain the complexity of the game and are sufficient for all 10 Zendo rules.[4]

The data is licensed under CC-BY 4.0.

**ActiveACRE.** We convert the originally visual tasks into symbolic version of the tasks, similar to [9]. While the ground truth rule always has the structure that the blicket machine produces noises when one or more "blicket" objects (each object is either a blicket or a non-blicket) is placed on the machine, in contrast to [9], we do not hint the learners that the ground truth rule is of this form, which means the learners are free to think that the rule may have to do with colors, number of objects, etc. We further modify the task to incorporate elements of active learning, making the logistics similar to Zendo: the game starts with 8 relevant objects, described with color (gray/red/blue/green/brown/cyan/purple/yellow), material (metal/rubber), and shape attributes (cube/sphere/cylinder), placed on the blicket machine which causes the machine makes sounds and follows by 7 rounds of query. The prediction phase tests the models on all possible combinations of the eight objects. We call this resulting domain, ActiveACRE. Figure 1 partially shows what a gameplay of simplified ActiveACRE looks like.

To obtain the 8 initial objects, we sample uniformly from the three attributes to get an object and keep doing this until we achieve 8 unique objects. This can be done with a simple code, without external data.

### A.4 Algorithm details

For all methods, unless specified, the number of LLM calls used per each iteration is 5 with each call batch-sampling 15 natural language hypotheses.

**Batch Inference.** For batch, fuzzy model, we cap the number of unique hypotheses considered to 30, otherwise we would have too many hypotheses considered, since all fuzzy hypotheses have non-zero posterior probability, making inference very compute intensive.

**Batch Inference with Refinement.** We set the number of refinement to 2 (we have tried increasing the number of refinement to 4 but didn't see any improvement). Following [9], this method works as follows: (1) it first batch-samples many hypotheses with LLM, (2) select the best hypothesis (in

---

[4]The 10 rules we use are "there's a red", "all are the same size", "nothing is upright", "one is blue", "there's a small blue", "all are blue or small", "a red is bigger than all non reds", "some touch", "a blue and a red touch" "some pieces are stacked", and "some pieces are stacked"

---

**Algorithm 1** LLM-SMC-S algorithm

---

Let $(x_1, y_1)$ be the first data point we observe

$h_1^{(1)}, ..., h_1^{(n)} \sim q(h|x_1, y_1)$

$w_1^{(i)} \leftarrow \frac{p(x_1, y_1, h_1^{(i)})}{q(h|x_1, y_1)}$ for $1 \le i \le n$        ▷ Reweighting

$H_1 \leftarrow Resampling(H_1, W_1)$        ▷ Resampling

**for** $t = 2, ..., T$ **do**

     The active learning algorithm gives $(x_t, y_t)$

     $h_t^{(1)}, ..., h_t^{(n)} \sim q(h|H_{t-1}, x_{1:t}, y_{1:t})$        ▷ Rejuvenating

     $A(h_t^{(i)}, H_{t-1}, W_{t-1}) = \frac{1}{n} \sum_{j=1}^{n} w_{t-1}^{(j)} \frac{p(h_t^{(i)}|x_{1:t}, y_{1:t}) r(h_{t-1}^{(j)}|h_t^{(i)}, x_{1:t}, y_{1:t})}{p(h_t^{(j)}|x_{1:t-1}, y_{1:t-1})}$

     $w_t^{(i)} \leftarrow \frac{A(h_t^{(i)}, H_{t-1}, W_{t-1})}{q(h_t^{(i)}|H_{t-1}, x_{1:t}, y_{1:t})}$ for $1 \le i \le n$        ▷ Reweighting

     $H_t \leftarrow Resampling(H_t, W_t)$        ▷ Resampling

**end for**

---

**Algorithm 2** B function pseudocode

---

**function** $\text{B}(x_{1:t}, y_{1:t}, H)$

     $result = \emptyset$

     $h_1, ..., h_k = \text{top-k-lowest-likelihood}(H, x_{1:t}, y_{1:t})$   ▷ get $k$ hypotheses with lowest likelihood

     **for** $i = 1, ..., k$ **do**

         $H_{nb} = LLM(x_t, y_t, h_i)$        ▷ get neighbors (nb) of $h_i$

         $H_{nb} = \{h_{nb} \in H_{nb} \mid p(h_{nb}|x_{1:t}, y_{1:t}) \le p(h_i|x_{1:t}, y_{1:t})\}$      ▷ filter out bad neighbors

         **if** $|H_{nb}| >$ m **then**        ▷ we want to consider a maximum of $m$ neighbors

            $w_{nb}^{(i)} \leftarrow p(h_{nb}^{(i)}|x_{1:t}, y_{1:t})$ for $1 \le i \le n$

            $H_{nb} \leftarrow Down-Sampling(H_{nb}, p = W_{nb}, size = m)$

         **end if**

         $result = result \cup H_{nb}$

     **end for**

      **return** $result$

**end function**

---

numbers of data points accounted) to be refined, (3) use LLM to output a batch of refined hypotheses, and (4) repeat the (2)-(3) steps until at least one hypothesis fully accounts for all data points.

**Online Inference (LLM-SMC-S)** The algorithm for LLM-SMC-S is described in Algorithm 1. For the first iteration, the initial important proposer $q(h|x_1, y_1)$ is defined to be an LLM, similar to batch inference. We define the forward kernel $q$ in the algorithm as follows:

$$q(h|H, x_{1:t}, y_{1:t}) \propto \mathbf{1}[h \in (H \cup B(x_{1:t}, y_{1:t}, H))] \tag{19}$$

The pseudocode for $B$ can be founded at Algorithm 2. What $B$ is doing is basically look at low likelihood hypotheses, prompt LLM to come up with their neighbors, and filter out bad neighbors and limit the number of chosen neighbors to $m$. We find that having the down-sampling step to keep the number of neighbors considered low is helpful in practice, but one can remove this step to make $B$ fully deterministic. The LLM function in the pseudocode means prompting an LLM with zero temperature.

### A.5 Example hypothesis traces from models

**Batch.** 'Blocks must touch at least one other block' is proposed but is immediately falsified by an existing experiment where a scene with no blocks touching is negative.

**Batch with refinement.** 'Blue blocks must not touch green blocks' is proposed and then refined into 'Blue blocks must not touch blocks of any color other than red'. This hypothesis later gets falsified, without an opportunity to refine itself since the model is not online, when a scene with no blocks touching is negative.

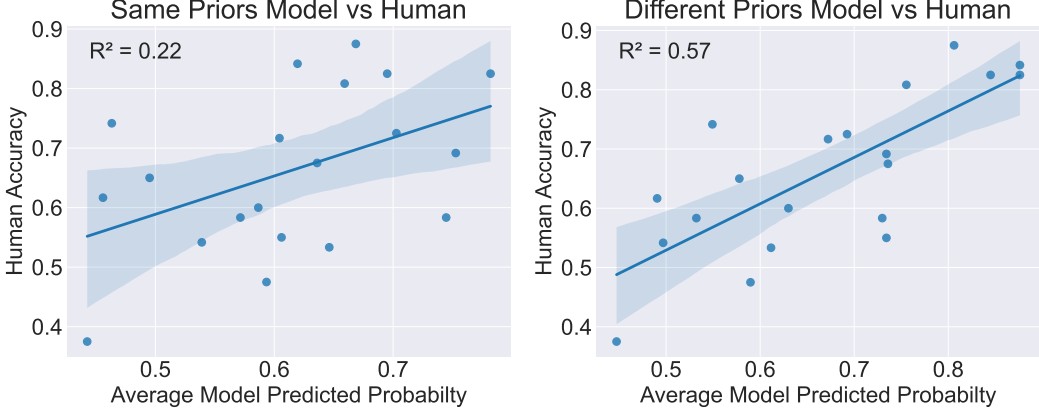

Figure 8: Human vs online, fuzzy model accuracy binned by 4 rule-following (RF) and 4 not rule-following (Not RF) test scenes. This figure shows online, fuzzy model with same and different priors for $\epsilon$ and $\delta$

**Online.** 'There must be a blue block' is proposed and added to the pool of particles. Since it has higher prior than other particles (has shorter length); it keeps surviving while others get killed, despite some conforming with the data. Upon seeing a scene with a blue touching a green being negative, the particle 'There must be a blue block' is perturbed into 'there must be a blue block touching a red block'.

## A.6  Baseline descriptions

In probabilistic inference terms, both batch with and without refinement correspond to importance sampling $p(h|x_{1:t}, y_{1:t}) = E_{p(h'|x_{1:t},y_{1:t})}[1[h = h']] = E_{q(h'|x_{1:t},y_{1:t})}[\frac{p(h'|x_{1:t},y_{1:t})}{q(h'|x_{1:t},y_{1:t})}1[h = h']]$.

The difference in the two baselines lie in how $q(h'|x_{1:t}, y_{1:t})$ is constructed.

**Batch.** $q(h|x_{1:t}, y_{1:t}) = U(LLM(x_{1:t}, y_{1:t}))$ where $LLM(...)$ prompts an LLM to return a list of hypotheses

**Batch with refinement.** $q(h|x_{1:t}, y_{1:t}) = U(Refined\text{-}LLM(x_{1:t}, y_{1:t}, None, 0))$ where $Refined\text{-}LLM$ is defined as follows:

First, let $s(h, x_{1:t}, y_{1:t}) = \frac{1}{t}\sum_{i=1}^{t} \mathbb{1}[h(x_i) = y_i]$. This simply scores what percentage of data points in $x_{1:t}, y_{1:t}$ that $h$ makes correct predictions. Then,

function $Refined\text{-}LLM(x_{1:t}, y_{1:t}, h, k)$:

    $H = LLM\text{-}with\text{-}h(x_{1:t}, y_{1:t}, h)$ # Prompts LLM to refine h

    if $k = K$:

        return $\emptyset$

    else if $\exists h' \in H, s(h', x_{1:t}, y_{1:t}) = 1$:

        return $h' \in H|s(h', x_{1:t}, y_{1:t}) = 1$

    else:

        $h^* = argmax_{h' \in H}(s(h', x_{1:t}, y_{1:t}))$

        return $Refined - LLM(x_{1:t}, y_{1:t}, h^*, k + 1)$

where $K$ is the number of refinements allowed.

| Method | Number of LLM Calls Per Iteration | | |
|---|---|---|---|
| | 1 | 5 | 10 |
| Batch, Fuzzy | $4.58 \pm 0.12$ | $4.57 \pm 0.15$ | $4.56 \pm 0.16$ |
| Online, Fuzzy | $5.11 \pm 0.04$ | $5.35 \pm 0.09$ | $5.28 \pm 0.06$ |
| Batch, Hard | $6.16 \pm 0.17$ | $6.01 \pm 0.19$ | $6.18 \pm 0.14$ |
| Batch w/ Refinement, Hard | $6.15 \pm 0.16$ | $6.18 \pm 0.14$ | $5.83 \pm 0.16$ |
| Online, Hard | $6.15 \pm 0.26$ | $\mathbf{6.55 \pm 0.13}$ | $\mathbf{6.38 \pm 0.11}$ |

Table 5: Average predictive posterior (standard error computed over 5 seeds) of models with different number of LLM calls (each LLM batch-samples 15 hypotheses) on Zendo.

## A.7 Priors

**Priors for $\epsilon$ and $\delta$**   $p(\delta)$ has a mean of 0.7 and a standard deviation of 0.1, and $p(\epsilon)$ has a mean of 0.9 and a standard deviation of 0.01. Both distributions are truncated to remain within the range [0.5, 1]. We found that using different priors for $\epsilon$ and $\delta$ results in a more human-like behavior as shown in Figure 8.

**Prior for $h$**   We let $p(h)$ be inversely proportional to the word count of $h$ for Zendo and uniform for ActiveACRE.

For Zendo, we consider using a prior that would decay exponentially in length but find that letting $p(h) \propto (\frac{1}{word\_count(h)})^2$ already makes the particles become mostly short strings. A prior decaying exponentially in string length would definitely be too harsh on the hypotheses.

## A.8 Computational cost

**Computational Cost Analysis**   Table 5 shows the performance of models with different compute budgets (number of LLM calls per iteration) on Zendo. It turns out that the performance of batch inference models plateaus after just 15 hypotheses (1 LLM call), while the performance of online inference models benefits from being able to sample more hypotheses but also plateaus after 75 hypotheses (5 LLM calls).

**Experiments Compute Resources**   We also describe here the compute resources required to reproduce the experiments. The main compute cost comes from OpenAI API which we call to prompt GPTs. The models with 1, 5, 10 LLM calls per each iteration uses up roughly $0.5, $1.5, $3 OpenAI API credit to run a Zendo task. For Zendo, one needs to run 50 tasks—10 Zendo tasks on 5 different seeds—to get the performance numbers of a method like we reported. The actual cost, however, could be lower than calculated since one can cache LLM responses.

## A.9 Human study on 'majority is red' rule details

20 participants from our academic department were recruited via Slack to attempt the rule "the majority of blocks are red". The participants are compensated $10-$20 depending on their performance ($10 base rate + $1.25 bonus for each correct test scence prediction – there are 8 test scenes). Figure 9 shows the web interface displayed to participants. The full instructions given to human participants are displayed at Figures 12 and 13.

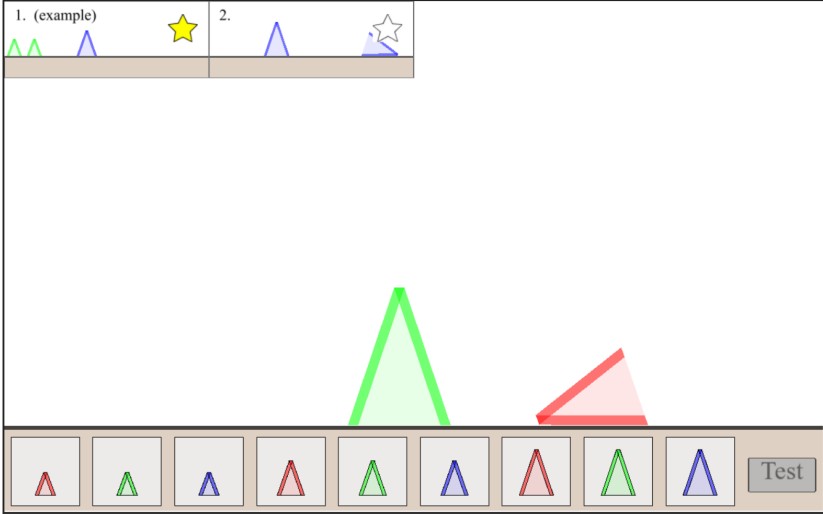

Now you play with the **bemmies**. Press buttons at the bottom to add blocks.
• Move the blocks around by picking them up with the mouse (left clicking and holding)
• Turn them using the "Z" (counterclockwise) and "X" (clockwise) keys
• Right click on them to remove them (command + click if you are using mac trackpad).
When you're done moving the special blocks, click "Test" to see if stars will come out

Figure 9: Example of the web interface shown to participants.

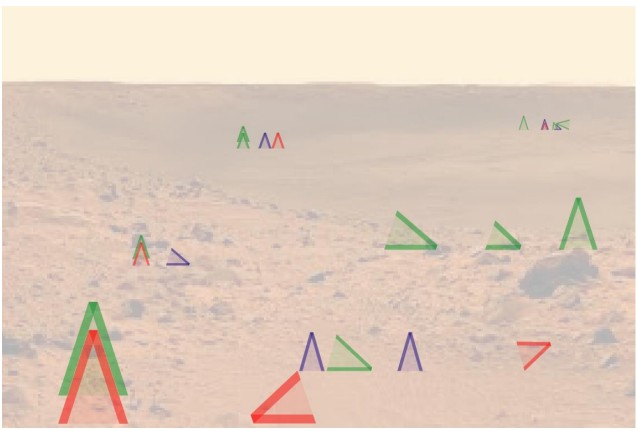

Figure 10: First figure for human participants instructions shown at Figure 12

1. These special blocks are called Blickets. Stars come out of them if **there is at least one small block**

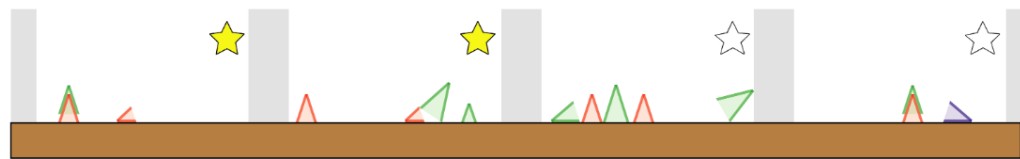

2. These special blocks are called Wozzles. Stars come out of them if **the blocks are all the same color**

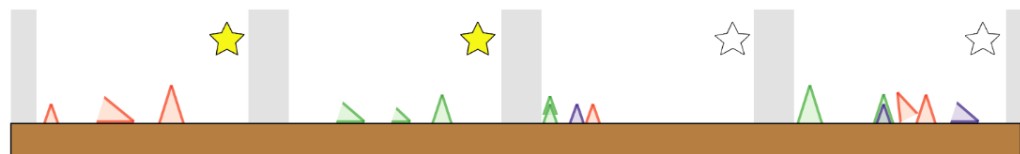

3. These special blocks are called Daxes. Stars come out of them if **none of the blocks are touching**

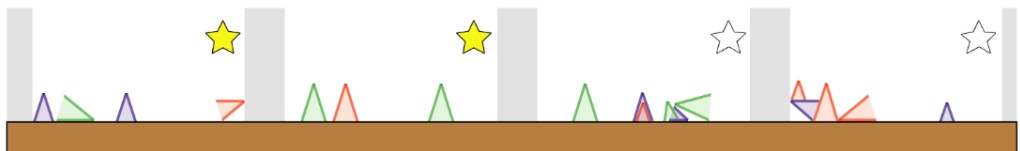

4. These special blocks are called Timas. Stars come out of them if **all the blocks point in the same direction**

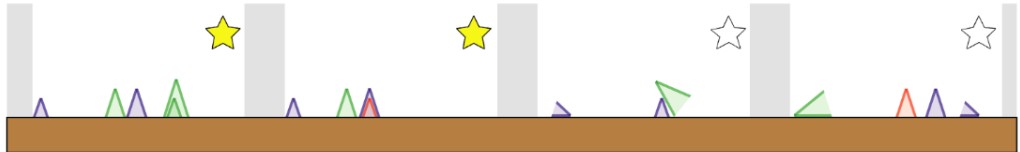

Figure 11: Second figure for human participants instructions shown at Figure 12

Thank you for playing my game!

In this game you will be learning about an alien planet. This planet is called Zorb.

On Zorb, there are these special blocks that look like this:

{Figure 9}

These special blocks may look the same, but there are many different kinds of special blocks on the planet Zorb. They all have different names, and they all work differently.

Sometimes, when the blocks are set up in certain ways, stars will shoot out of them! Every kind of special block has a different rule for making stars shoot out of them.

Your job is to figure out each rule for how to make stars come out of all of the different kinds of special blocks!

{Figure 10}

So there are a lot of things that might make stars come out of certain special blocks!

You might get stars from blocks of different numbers, from blocks of different colors, from blocks of different sizes, from blocks facing different directions, and more!

We found out that there are 2 more kinds of special blocks on Zorb! Let's call them 'Bemmies' and 'Yoks'. Again, you do not have to memorize the names -- we just want to emphasize that different kinds of blocks work under different rules

But we don't know the rule for setting up each kind of special blocks so stars will come out of them. Your job is to figure out the two different rules for how to set up each different kind of special blocks so stars come out!

Now we're going to watch a video. This video is going to show you how you can move the special blocks around yourself!
You must watch the video to continue.

So, in the interface you can:
    Press buttons at the bottom to add blocks
    Move the blocks around by picking them up with the mouse (left clicking and holding)
    Turn them using the "Z" (counterclockwise) and "X" (clockwise) keys
    Right click on them to remove them (command + click if you are using mac trackpad)

When you're done moving the special blocks, you're going to test them to see if stars will come out of them. If you set them up in the right way according to the rule, you'll see a bunch of stars appear! Otherwise, nothing will happen.

Figure 12: (Part 1) Instructions for participants. Please find instruction figure 1 and 2 at Figures 10 and 11

After you move the special blocks around and test them, you're going to see if you can pick out which pictures of the blocks you think will shoot out stars. This video will show you how to do that:

{demo video}

In the video you can see that this participant thinks that four of the pictures show bemmies that stars will come out of (the ones marked in grey). The right answer could be anywhere between one and seven of the pictures.

Adults: You will earn a bonus of $1.25 for each of the pictures in the main task where you guess correctly whether it will shoot out stars (demo task performance does not count). That means, if you get all eight pictures correct in the main task, you will earn a bonus of $10!

You must watch the video to continue.

{demo video}

Finally, you may guess the rule for how this kind of special blocks works.

For example, if it looks like stars only shoot out of the blocks if all of them are green, you would write something like: "all the blocks have to be green"!

Warning: Your responses will be checked by a human before HIT approval. Nonsensical or copy-pasted answers will lead to your HIT being rejected. If you truly have no ideas about a rule, please just write "I do not know".

Instructions Summary:
    You will look at 2 different kinds of special blocks (including one demo task for learning the game) that will shoot out stars if they are set up in certain ways.

    You must figure out the rule for how each kind of special blocks works.

    You will set up the special blocks and test them to see if stars will shoot out of them seven times for each type.

    Your goal is to figure out which out of 8 new pictures of each kind of special blocks will shoot out stars ($1.25 bonus for each correct in the main task)...

    ...and to write down your best guess of the rule for that kind of special block!

Figure 13: (Part 2) Instructions for participants.

### A.10 Prompts

The prompts used in all of our experiments can be found in Tables 7 to 12.

For Zendo, we engineer the prompts for initial importance sampler $q(h|x, y)$ for online inference so that they only output simple rules (see Table 7); this approach helps the proposer output hypotheses with higher priors, since our prior is defined by the number of words in the rule. We cannot apply this trick to batch inference because, unlike online inference, it does not evolve simpler rules into more complex ones. Additionally, we also design the importance sampler prompts to avoid proposing negative rules ('there is no ...') (see Table 7). We found that this leads to a more human-like behavior and also better performance.

### A.11 Supplemental results

See Table 6 and figs. 14 and 15

| Method | LogL |
|---|---|
| [13]'s best model | 3085 |
| Batch, Fuzzy | 3352.93 |
| Online, Fuzzy | **2988.77** |
| Batch, Hard | 5842.00 |
| Batch w/ Refinement, Hard | 6999.52 |
| Online, Hard | 10419.86 |

Table 6: BIC scores of models on human data.

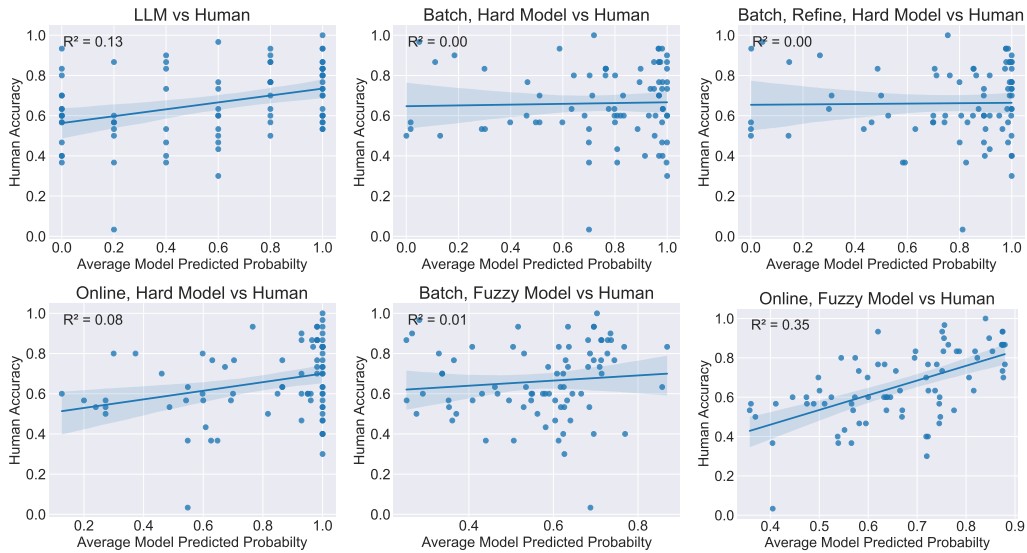

Figure 14: Comparing human and model prediction on each test scene after 7 rounds of experimentation. Each point is a prediction on a test scene.

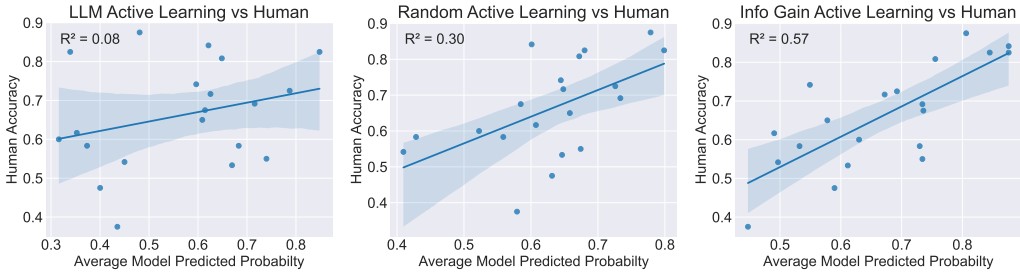

Figure 15: Human vs online, fuzzy model accuracy binned by 4 rule-following (RF) and 4 not rule-following (Not RF) test scenes. This figure shows online, fuzzy model with three different active learning methods: LLM, Random, and InfoGain

| Method | Prompts for Zendo | Prompts for ActiveACRE |
|---|---|---|
| Batch Inference | Given the following structures described with {att_summary} of blocks in the structures: {text_c} Please list {num} possible rules about the attributes in a structure that differentiate the good structures from the bad structures. Keep in mind that 1. All bad structures must violate the rules. 2. Orders of blocks in a structure do NOT matter. 3. Do NOT propose "negative rules" such as "there is no green block". 4. The rules are short, concise, single sentences. Please number them from 1-{num} and do not say anything else | A group of objects may make the "blicket machine" have lights turned on or off depending on the objects in it. We seek to figure out the rule underlying this. Consider the following: {text_c} Please state {num} possible rules what makes the light turned on. State them in a listed number. Do not explain. |
| Online Inference | Please list {num} possible rules about the {att}. Example 1: Structure: blue, blue Simple rules (Orders do NOT matter): 1. There is a blue block 2. All blocks are blue Do NOT propose "negative rules" such as "there is no green block". Do NOT propose rules with quantifier such as "there are two blue blocks" Task 1: {x} Simple rules (Orders do NOT matter): | A group of objects may make the "blicket machine" have lights turned on or off depending on the objects in it. We seek to figure out the rule underlying this. Consider the following: {text_c} Please state {num} possible rules what makes the light turned on. State them in a listed number. Do not explain. |

Table 7: Prompts used for the importance sampler $q(h|x_1, y_1)$ of all methods

| Prompts for Zendo | Prompts for ActiveACRE |
|---|---|
| A structure has one or more blocks. Each block should contain the following attributes:
{att_par}

Example of rule modifications:
Quantifier change: 'There must be a green block' -> 'There are two green blocks'
Additional attribute: 'There must be a green block' -> 'There must be a green block that is upright'
Attribute change: 'There must be a green block' -> 'There must be a blue block'
These modifications are "local": only one attribute/quantifier is changed or added for each modification.

Please modify the rule '{h}'. Generate {num} rules for each type of modification (Quantifier change, Additional attribute, Attribute change) so that the following structure is {text_y} a good structure:
{x}
Note that the number of the blocks do not matter.

Make the format a numbered list (1., 2., ..., 15.) Remember that the new rules should be a "local" modification from the rule '{h}'. Do not use attribute values that are not mentioned earlier. Do not say anything other than the modified rules. | An object contains the following attributes:
color (gray/red/blue/green/\ brown/cyan/purple/yellow)
material (metal/rubber)
shape(cube/sphere/cylinder)

Example of rule modifications:
Additional conjunction: 'The light turns on when there is a cylinder present' -> 'The light turns on when there is a cylinder and a cube present'
Additional disjunction: 'The light turns on when there is a cylinder present' -> 'The light turns on when there is a cylinder or a cube present'
Additional attribute: 'The light turns on when there is a cylinder present' -> 'The light turns on when there is a blue cylinder present'
These modifications are "local": only one disjunction/conjunction/attribute is changed or added for each modification.

Please modify the rule '{h}'. Generate {num} rules for each type of modification (Additional conjunction, Additional disjunction, Additional attribute) so that the light does {text_y} turn on when the following objects are present:
{x}
Note that the number of the blocks do not matter.

Make the format a numbered list (1., 2., ..., 15.) Remember that the new rules should be a "local" modification from the rule '{h}'. Do not use attribute values that are not mentioned earlier. Do not say anything other than the modified rules. |

Table 8: Prompts used for the forward kernel $q(h'|H_t, x_{1:t}, y_{1:t})$ of online inference methods

| Prompts for Zendo | Prompts for ActiveACRE |
|---|---|
| Given the rule '{h}', please give one structure that conforms with the rule and another structure that violates with the rule.

A structure has one of more blocks. Each block should contain the following attributes: {spec}{stacking_note}

The format of each structure should be as follows:
(conforms with the rule) Structure 1: {example_block}

(violates the rule) Structure 2: {example_block} | Given the rule '{h}', please give one group of objects that makes the light turned on and another that makes the light turned off

The list of available of objects are {all_objects}.

The format of your answer should as follows:

light on group of objects: obj_1, obj_2, ...

light off group of objects: obj_1, obj_2, ...

All objects in a group must be unique. Do not say anything else. |

Table 9: Prompts used for experiment proposers.

| Prompts for Zendo | Prompts for ActiveACRE |
|---|---|
| Please synthesize a python program that implements the rule '{h}'

The program should takes in a ZendoStructure which represents a structure and returns True if it's a good structure and False otherwise.

The docstrings for the classes are as follow:

class ZendoStructure:
    :param blocks: list of ZendoBlock

class ZendoBlock:
    :param color: str (blue/red/green)
    :param size: str (small/medium/large)
    :param orientation: str (upright/left/right/strange)
    {groundedness_param_msg}
    :param touching: list of int (index starts at 1)

The signature for the synthesized program should be
def rule(structure: ZendoStructure) -> bool

Only output the 'rule' function. Do not include anything else. | Please synthesize a python program that implements the rule '{h}'

The program should takes in a ACREGroup which represents a group of objects and returns True if it's a good group and False otherwise.

The docstrings for the classes are as follow:

class ACREGroup:
    :param objs: list of ACREObject

class ACREObject:
    :param color: str (gray, red, blue, green, brown, cyan, purple, yellow)
    :param material: str (metal, rubber)
    :param shape: str (cube, sphere, cylinder)

The signature for the synthesized program should be
def rule(group: ACREGroup) -> bool

Only output the 'rule' function. Do not include anything else. |

Table 10: Prompts used to translate natural language $h$ to code.

| Prompts for Zendo | Prompts for ActiveACRE |
|---|---|
| A structure has one or more blocks. Each block should contain the following attributes: {att_par} | An object contains the following attributes: color (gray/red/blue/green/\ brown/cyan/purple/yellow) |
| Consider the following rule: '{h}' | material (metal/rubber) shape(cube/sphere/cylinder) |
| Given a structure, the output is yes if it follows the rule (or is a good structure) and no if it does not (or is a bad structure) | Consider the following rule: '{h}' |
| The given rule gives incorrect output for the following structures: | The given rule gives incorrect output for the following groups of objects: |
| {feedback} | {feedback} |
| Based on the given rule, generate {num} new refined rules that fix the outputs for all mentioned structures. The new rules may involve any of the mentioned attributes (color, size, orientation, grounded, touching). Please number them from 1-{num} and do not say anything else | Based on the given rule, generate {num} new refined rules that fix the outputs for all mentioned structures. Please number them from 1-{num} and do not say anything else |

Table 11: Prompts used to perform refinement in batch inference with refinement

| | Prompts for Zendo | Prompts for ActiveACRE |
|---|---|---|
| Initial prompt | You are playing an inductive game with me. I'll be the moderator, and your task is to figure out the secret rule that I know by coming up with a structure of blocks to ask me whether it conforms with the secret rule or not.

The structure has one of more blocks. Each block should contain the following attributes: {att_par}

To give you a start, I'll describe one structure that follows the rule:

{text_c}

Give a very short summary on what you currently think the secret rule is. | You are playing an inductive game with me. I'll be the moderator, and your task is to figure out the secret rule that I know by coming up with a group of blocks to ask me whether the group conforms with the secret rule or not.

An object contains the following attributes: color (gray/red/blue/green/\ brown/cyan/purple/yellow) material (metal/rubber) shape(cube/sphere/cylinder) The list of available of objects are {all_objects}.

To give you a start, I'll describe one group of objects that follows the rule:

{text_c}

Give a very short summary on what you currently think the secret rule is. |
| Follow-up prompt | The verdict on whether the queried structure follows the rule is {verdict}. Give a very short summary on what you currently think the secret rule is. | The verdict on whether the queried structure follows the rule is {verdict}. Give a very short summary on what you currently think the secret rule is. |
| Active learning prompt | Give one structure you want to test whether it follows the secret rule or not. Do not include anything other than the structure. | Give one structure you want to test whether it follows the secret rule or not. Do not include anything other than the structure. |
| Prediction prompt | Now, do you think this structure follow the rule?\n: {x}\nAnswer only yes or no. Give your best guess even if you are uncertain. Do not explain. Just say yes or no | Now, do you think this group of objects follow the rule?\n: {x}\nAnswer only yes or no. Give your best guess even if you are uncertain. Do not explain. Just say yes or no' |

Table 12: Prompts used for vanilla, direct LLM method

