# OpenReview forum: "Doing Experiments and Revising Rules with Natural Language and Probabilistic Reasoning"
_NeurIPS.cc/2024/Conference — NeurIPS 2024 poster_

### Official Review · Reviewer_zQ3i · 2024-06-16

**Soundness:** 2
**Presentation:** 2
**Contribution:** 2
**Rating:** 4
**Confidence:** 4

**Summary:**

This paper integrates a Large Language Model (LLM) within a Sequential Monte Carlo (SMC) algorithm, where the LLM functions as both the proposal distribution (revising existing particles) and the evaluator (assessing each hypothesis in light of new data). The authors applied this method to two cognitive psychology tasks: Zendo and ActiveACRE. These tasks involve formulating latent game rules, updating the probability of these rules based on new data, and proposing new experiments to gather further data.

In both domains, the Online, Hard variant of SMC consistently outperforms other SMC variants in identifying the best latent rule, while the Online, Fuzzy variant of SMC more accurately captures human data. However, compared to optimal experimentation (the InfoGain algorithm), none of the LLM-based proposers outperform the InfoGain algorithm. Furthermore, a random proposer outperforms the LLM-based methods, indicating a significant deficit in the LLM’s performance as a proposer.

**Strengths:**

I find the proposed method interesting, as it effectively demonstrates the benefits of integrating Bayesian methods with LLMs. The comparison of various LLM-based methods for problem solving and explaining human data is particularly valuable.

**Weaknesses:**

The main weakness of the work is that both the enhancement in problem-solving capability and the correspondence with human data are not convincing. While the Online, Hard variant of LLM-based SMC clearly outperforms ReAct-style LLMs, it employs LLMs in three distinct roles: as a proposer (for revising existing hypotheses), as an evaluator (for translating natural language hypotheses into code and verifying them), and as an active learner (for proposing new experiments to perform next). The performance of the LLM as an active learner, however, is particularly weak, even underperforming a random active learner (see Tables 3 and 4). Replacing the active learner in the Online, Hard method with a random active learner would likely significantly improve performance compared to using the LLM. Consequently, it is unclear which aspects of the pipeline benefit from the inclusion of the LLM and which aspects are detrimental.

Second, the comparison between humans and models is superficial. The best-fitting Online, Fuzzy model clearly includes at least two free parameters, as detailed in Section 2.3. However, the authors did not report the values of these best-fitting parameters. Are these the only two free parameters? Did you add additional decision noise to the model to better capture human data? If so, it would be more informative to report scores such as BIC or cross-validation results rather than just the simple log likelihood of the model.

Figure 7 aims to illustrate a bounded rationality model for human-likeness, suggesting that the Online, Fuzzy method best aligns with human behavior when the number of LLM calls per iteration is neither too few nor too many. However, the evidence for this claim appears weak. The bounded rationality pattern only emerges with the Online, Fuzzy method, and the authors tested only three different numbers of LLM calls per iteration (x-axis). With just three data points, it is difficult to conclusively demonstrate an inverted U-shaped relationship. Additionally, when LLMs were used with methods other than the Online, Fuzzy method, the inverted U-shaped relationship did not appear at all.

**Questions:**

Lines 227-229: The authors claim that “Indeed, Table 2 shows that our best-performing model surpasses [13]’s model on human data […], while our model does not perform such parameter fitting.” However, the Online, Fuzzy method does contain free parameters (specifically, the two free parameters that determine the false-positive and false-negative rates for each hard rule) that can be adjusted to better fit human data. Did the authors actually fit these parameters using human data?

**Limitations:**

I believe the paper would be improved if the authors could independently comment on the three separate roles of the LLMs (i.e., proposer, evaluator, and active learner) in their proposed method.

Minor comments:
Line 224: “judgement” —> “judgment”

---

> ### Author Rebuttal · Authors · 2024-08-05
>
> We thank the reviewer for the detailed feedback!
>
> > Concern 1: The main weakness of the work is that both the enhancement in problem-solving capability and the correspondence with human data are not convincing.  It is unclear which aspects of the pipeline benefit from the inclusion of the LLM and which aspects are detrimental.
>
> We would like to respond to this concern with two key points:
> 1. LLMs are necessary for natural language hypothesis generation:
> As discussed in the introduction, since the hypotheses are natural language, we need to leverage LLM in our proposer. Without LLMs, having natural language hypotheses would not be possible. We also compare our proposers against a non-LLM proposer of non-linguistic hypotheses from [1] in Table 2.
> 2. Our experiments are controlled:
> While our work leverages LLM in three distinct components: the hypothesis proposer, natural-language-to-code transpiler, and active learner, we control our experiments by keeping the transpiler and active learner fixed for all methods in the main results to perform an isolated study on the hypothesis proposer and keeping the transpiler and hypothesis proposer fixed in the active learning ablation study (Table 3 and 4).
>
> We address other points raised under concern 1 below.
>
> > Concern 1.1: While the Online, Hard variant of LLM-based SMC clearly outperforms ReAct-style LLMs, it employs LLMs in three distinct roles: as a proposer, as an evaluator, and as an active learner.
>
> In the main results (Table 1 and 2, and Figure 4) where we show that the online models clearly outperform ReAct-style LLMs in both accuracy and human-likeness, all methods use the same transpiler and active learner.
>
> > Concern 1.2: The performance of the LLM as an active learner, however, is particularly weak, even underperforming a random active learner (see Table 3 and 4). Replacing the active learner in the Online, Hard method with a random active learner would likely significantly improve performance compared to using the LLM.
>
> We believe there is a misunderstanding here. There are three different active learners: LLM, random, and InfoGain. InfoGain (introduced in section 2.2) involves choosing an experiment that maximizes information gain from a pool of candidate experiments proposed by the experiment proposer, which can either be LLM experiment proposer or random experiment proposer (an ablation study for the experiment proposer is done in table 4).
>
> Our work advocates for the InfoGain active learner which outperforms both LLM active learner and random active learner as shown in Table 3.
>
> All results, except for Table 3 and 4, use InfoGain with LLM experiment proposer, which is the best active learner, as its active learner.
>
> > Concern 2:  The comparison between humans and models is superficial.
>
> We believe this concern is caused by some misunderstandings. We clarify the misunderstandings below.
>
> > Concern 2.1:  The best-fitting Online, Fuzzy model clearly includes at least two free parameters, as detailed in Section 2.3. However, the authors did not report the values of these best-fitting parameters.
>
> We would like to first clarify that the two noise parameters, $\theta = (\epsilon, \delta)$ are not free parameters; they are random variables, as discussed in section 2 and section 2.3. Since they are random variables with priors on them, they are not optimized but rather get marginalized out as shown in Equation 2 and 3.
>
> However, the gaussian priors for the two noise random variables do have free parameters, namely the means and standard deviations. Thus, there are 4 free parameters in our fuzzy models and no free parameters in our hard models.
>
> Although in principle we could have fitted these 4 parameters, in practice, it is really expensive to do so because it involves lots of LLM calls to even evaluate a single parameter setting. Therefore, we did not fit the free parameters; we only tried two different parameter settings for all fuzzy models, and reported the results of the setting that worked best in the main paper and the results of the other parameter setting in the appendix (Figure 8). The same parameter setting is used across all fuzzy models and reported in the appendix (A.4).
>
> > Concern 2.2: It would be more informative to report scores such as BIC
>
> We include here a table of BIC scores which we will include in our revised appendix:
>
> | Method  | BIC      |
> |---|---|
> | [1]’s best model            | 3085     |
> | Batch, Fuzzy                | 3352.93  |
> | Online, Fuzzy               | **2988.77** |
> | Batch, Hard                 | 5842.00  |
> | Batch w/ Refinement, Hard   | 6999.52  |
> | Online, Hard               | 10419.86 |
>
> > Concern 3:  The evidence for bounded rationality appears weak. When LLMs were used with methods other than the Online, Fuzzy method, the inverted U-shaped relationship did not appear at all.
>
> The data provided in Figure 7 is meant to show that our most human-like model is consistent with the theory of bounded rationality, and we leave it for future study to thoroughly examine whether, or the degree to which, the model is boundedly rational.
>
> However, the fact that only our most human-like model, the online, fuzzy model, is the only model exhibiting the relationship is not a weakness; it is in fact the opposite: it helps strengthen our point. Specifically, we expect the inverted U-shaped relationship to appear only in models that sufficiently explain human data. We believe that other methods do not exhibit the inverted U-shaped relationship because they are not human-like enough to have the bounded rationality property.
>
> > Question 1: Contradicting to lines 227-229, the Online, Fuzzy method does contain free parameters that can be adjusted to better fit human data. Did the authors actually fit these parameters using human data?
>
> We refer the reviewer to our replies to concern 2.1 which addresses this question.
>
> [1] Bramley et al. 2018, Grounding compositional hypothesis generation in specific instances.

---

### Official Review · Reviewer_nwGu · 2024-07-12

**Soundness:** 3
**Presentation:** 3
**Contribution:** 3
**Rating:** 7
**Confidence:** 3

**Summary:**

The authors tackle the problem of learning natural language hypotheses and collecting new experiments to enable hypothesis refinement. They propose a system that integrates LLMs and SMC/BOED to solve this problem; crucially, this method goes beyond prior work on inductive reasoning by allowing for experimentation to guide hypothesis refinement. In brief, they use a LLM to propose natural language hypotheses, SMC to reweight and resample these hypotheses based on how well they explain the data, and then use SMC’s weighted particle approximation to approximate the expected information gain and select experiments.

The authors then evaluate this approach on two task Zendo and ActiveACRE, showing improvements against previous inductive reasoning methods and showing some agreement between their proposed system and human behavior on these tasks.

**Strengths:**

* The work presents a principled, novel approach to an interesting problem. First, this work extends previous work on inductive hypothesis generation with LLMs by accounting for experimentation. Second, I think the design choices are quite practical, interesting, and creative.

* I like the idea of considering natural language as the hypothesis space and using SMC and BOED for active experimentation/hypothesis revision is quite natural and well-motivated. SMC and BOED are well-established ideas but leveraging them for the task of natural language hypothesis generation and experimentation is new and the authors have solid approach to integrating LLMs for proposing hypotheses while still guaranteeing the correctness of their procedure.

* The authors ablate the different design decisions of the system.

* The presentation of the paper was mostly clear but I do have some suggestions.

**Weaknesses:**

I think the tasks do a fine job of evaluating the system but I'm not sure if they're the most compelling demonstrations of the value of natural language hypotheses/they might not fully utilize the power of natural language hypotheses. For example, it seems like a restricted DSL might be enough for some of these tasks; I do note that the authors do an analysis to show the benefit of natural language.

Additionally, there are a few ablations of the system that could strengthen the paper.

* Ablations that illustrate the role of a natural language hypothesis space or at least some additional discussion of more realistic settings where this is a good design choice.

* Number of particles/number of hypotheses

* Some analysis of particle degeneracy would be helpful

* How does a model with vision capabilities perform on this task (e.g., GPT-4 V)?

I think the presentation can be improved a bit in some places. For example, Figure 4b could probably be replaced with a summary figure. It’s a bit difficult to parse a 6x10 grid of figures.

Furthermore, it was sometimes hard to understand the details of some of the baselines (without reading the hypothesis refinement/search papers). For example, I think clearly defining the batched inference with refinement baseline (in math) would be helpful in the main text since it's a crucial method to understand.

**Questions:**

* Does LLM-SMC-S produce a consistent estimator of expectations of test functions wrt the normalized posterior (this is the guarantee enjoyed by vanilla SMC)? I think this probably follows from the properly weighted property and the fact that ratios of consistent estimators are consistent. It would be good if authors could comment on this in the paper though.
* Does the procedure suffer from particle degeneracy? It’s fine if it does but since this is a considerable practical challenge of SMC, I think it’s worth commenting on this.
* The derivation in A.1 has some typos. I think there’s a missing integral sign in line 10.
* I think I would have liked to see some of the actual hypotheses proposed by the LLM in the main text.
* I think it would be helpful to see Equation 6 is actually approximated explicitly in math.
* Figure 2 was a bit confusing for someone who’s used to seeing plots in SMC papers where the circles correspond to particles; here I think the circles represent specific natural language hypotheses (e.g., 3 red balls) but at first I thought they represented particles.

**Limitations:**

The authors discuss limitations.

---

> ### Author Rebuttal · Authors · 2024-08-05
>
> We thank the reviewer for the positive feedback! Below we address concerns and questions.
>
> > Concern 1: I'm not sure if they're the most compelling demonstrations of the value of natural language hypotheses/they might not fully utilize the power of natural language hypotheses. For example, it seems like a restricted DSL might be enough for some of these tasks
>
> We fully acknowledge this concern.
>
> Our work evaluates the models on traditional, well-studied Bayesian concept learning datasets to demonstrate that using natural language representation is beneficial even in domains where crafting a DSL is still possible and also to enable in-depth analyses.
>
> We believe that our work is needed to convince the Bayesian concept learning community to consider using natural language representation, and it is also an important stepping stone towards fully wielding the power of natural language.
>
> As the reviewer pointed out, we do show the almost obvious flexibility of natural language representation in our “majority of blocks are red” experiment. We leave evaluations on more complex domains where natural language representation is a must for future work.
>
> > Concern 2: There are a few ablations of the system that could strengthen the paper.
> - Ablations that illustrate the role of a natural language hypothesis space or at least some additional discussion of more realistic settings where this is a good design choice.
> - Number of particles/number of hypotheses
> - Some analysis of particle degeneracy would be helpful
> - How does a model with vision capabilities perform on this task (e.g., GPT-4 V)?
>
> Realistic setting: more abstract inductive reasoning domains such as content recommendation and moral reasoning from the GATE paper [1]
>
> Number of hypotheses: while not directly ablated, the number of hypotheses is indirectly tied to the number of LLM calls per iteration, which we performed an ablation study on
>
> Detailed analysis on LLM-SMC-S, which, we think, may be able to provide a probabilistic account of evolutionary algorithms, and extension to visual data through the use of multimodal LLM are both exciting future directions we plan to explore.
>
> > Concern 3: It was sometimes hard to understand the details of some of the baselines
>
> Please see our global response where we provide more details of the baselines. They will be included in our revised paper.
>
> > Question 1: Does LLM-SMC-S produce a consistent estimator of expectations of test functions wrt the normalized posterior (this is the guarantee enjoyed by vanilla SMC)?  It would be good if authors could comment on this in the paper though.
>
> Yes, it does. We will revise our paper accordingly.
>
> > Question 2: Does the procedure suffer from particle degeneracy? It’s fine if it does but since this is a considerable practical challenge of SMC, I think it’s worth commenting on this.
>
> From what we see during the experiments, particle degeneracy usually happens when the ground truth rule is discovered and included in the pool of particles. Since the ground truth rule particle has a much higher posterior than others, other particles tend to get killed. When the ground truth rule has not been discovered, the particles tend to still be diverse.
>
> > Question 3: The derivation in A.1 has some typos.
>
> Thank you for pointing it out! We will fix it.
>
> > Question 4: I think I would have liked to see some of the actual hypotheses proposed by the LLM in the main text.
>
> Please see our global response where we provide actual hypothesis traces from our models. They will be included in our revised paper.
>
>
>
> > Question 5: I think it would be helpful to see Equation 6 is actually approximated explicitly in math.
>
> Equation 6 is actually the approximation of Equation 4, and it can be computed exactly.
>
> > Question 6: Figure 2 was a bit confusing for someone who’s used to seeing plots in SMC papers where the circles correspond to particles
>
> The figure does in fact represent particles with circles. The three red particles represent how there are three particles with the same hypothesis. The colors of these circles change when their hypotheses are revised.
>
> [1] Li and Tamkin et al. 2023. Eliciting Human Preferences with Language Models.

---

> > ### Comment · Reviewer_nwGu · 2024-08-12
> > **Acknowledged and maintain my positive support**
> >
> > Thanks for your clarifications! I"ll maintain my score and advocate/support for the acceptance of this paper.
> >
> > I only have one presentation comment.
> >
> > For Proposition 1, consider making it a statement about consistency of LLM-SMC-S? Presumably, consistency is the guarantee that people care about and the properly weighted property is why this guarantee holds (I could be missing something though).
> >
> > Regarding, the author's response to the first concern. That's reasonable and consider elaborating further in the Discussion (although this is already briefly discussed).

---

### Official Review · Reviewer_C8N9 · 2024-07-13

**Soundness:** 3
**Presentation:** 3
**Contribution:** 3
**Rating:** 7
**Confidence:** 4

**Summary:**

The paper addresses the problem inferring rules and designing experiments based on them. To do so, 1) they propose representing rules in natural language, generated with LLMs. 2) Using Monte-Carlo algorithms to score them 3) Revising these rules and proposing new experiments with LLMs. They instantiate the problem in the domains of Zendo (binary rules) and ActiveAcre (abstract causal reasoning/ blicket detector style tasks). The authors then compare human inferences with LLM inferences to show what might explain human inferences and how their method can outperform humans in inferring underlying rules.

**Strengths:**

- The paper is well written, I enjoyed reading it!
- The problem is well motivated, and the description of the methods is also clear.
- The authors use good baselines to compare their method with. I liked the controls and conditions used.
- I particularly liked the rule following analysis and visualizations presented in fig 4b
- I also liked the discussion on using formal languages vs natural language to represent rules!

**Weaknesses:**

- I think the description of the baselines could be made clearer. Including the prompts in the main or using figures to describe them could be useful.
- While I liked the discussion on formal languages vs natural languages, I would have liked to see a baseline where the model implements rules in a formal language. This need not be a DSL, but a way for the model to reason with an external symbolic solver. Eg: the language could be python (with pymc/stan/pyro), or webppl.
    - Relatedly, [https://arxiv.org/abs/2402.17879](https://arxiv.org/pdf/2402.17879) seems like a relevant paper where rules are inferred and implemented in pymc
- Finally, the two domains, while inspired from psychology experiments, are quite artificial. They did make it possible for the authors to do an in-depth analysis (with human responses and try different variations, analyse failures) but adding a more realistic domain the strengthen the contribution of the paper.

**Questions:**

See weaknesses.

**Limitations:**

The authors address limitations.

---

> ### Author Rebuttal · Authors · 2024-08-05
>
> We thank the reviewer for the positive feedback! Below, we address concerns raised by the reviewer.
>
> > Concern 1: I think the description of the baselines could be made clearer.
>
> Please see our global response where we provide more details of the baselines. They will be included in our revised paper.
>
> > Concern 2:  While I liked the discussion on formal languages vs natural languages, I would have liked to see a baseline where the model implements rules in a formal language. This need not be a DSL, but a way for the model to reason with an external symbolic solver. Eg: the language could be python (with pymc/stan/pyro), or webppl.
>
> We did not consider such a baseline because:
>
> 1. A pair of recent papers found that pure python generation underperforms NL->Python (Ellis ‘23 NeurIPS [1], Wang ‘24 ICLR [2])
>
> 2. We compare with the Bramley et al. [3] Zendo model, which uses a formal language
>
> We will mention these facts in the revision.
>
> > Concern 3: The two domains, while inspired from psychology experiments, are quite artificial. They did make it possible for the authors to do an in-depth analysis, but adding a more realistic domain the strengthen the contribution of the paper.
>
> We fully acknowledge this concern.
>
> Our work evaluates the models on traditional, well-studied Bayesian concept learning datasets to demonstrate that using natural language representation is beneficial even in domains where crafting a DSL is still possible and also to enable in-depth analyses.
>
> We believe that our work is needed to convince the Bayesian concept learning community to consider using natural language representation, and it is also an important stepping stone towards fully wielding the power of natural language.
>
> We do show the almost obvious flexibility of natural language representation in our “majority of blocks are red” experiment. We leave evaluations on more complex domains where natural language representation is a must for future work.
>
>
> [1] Ellis 2023, Human-like Few-Shot Learning via Bayesian Reasoning over Natural Language.
>
> [2] Wang et al. 2024, Hypothesis Search: Inductive Reasoning with Language Models.
>
> [3] Bramley et al. 2018, Grounding compositional hypothesis generation in specific instances.

---

> > ### Comment · Reviewer_C8N9 · 2024-08-11
> >
> > Thank you for your responses and addressing my concerns! I'll keep my current score.

---

### Official Review · Reviewer_i693 · 2024-07-14

**Soundness:** 4
**Presentation:** 4
**Contribution:** 3
**Rating:** 7
**Confidence:** 4

**Summary:**

This paper describes a model of online construction of rules that explain data, where (a) hypotheses are expressed in natural language, and (b) they are updated by proposing experiments (inputs to the ground-truth rule). The method consists of an extension of SMC, where experiments and hypotheses revision are proposed by an LLMs; the best experiment is chosen by an approximate information gain criterion. The paper considers both fuzzy and hard rules, and finds that hard rules perform best in terms of task completion (and the model outperforms previous methods, especially those that propose a batch of hypotheses rather than inferring them online). Moreover, the fuzzy rules model best fits human behavior.

**Strengths:**

The paper is clear and well motivated. The setup of online inference, where the agent is allowed to propose experiments and observe results, is interesting and new in this line of work. Though the initial domains studied in this setup are simple, they allow us to understand the meat of the problem more clearly. I can imagine other more realistic domains, such as based on code debugging or data analysis, that might benefit from the ideas here.

The paper is careful in its choice of experiments. In Zendo and ActiveAcre, the paper compares with recent, relevant baselines: ReAct (interactive, but LLM only), batch inference (static), batch with refinement (as done in prior work), or online inference. It also accomplishes the goal of obtaining decent fit to human data, which previous related papers did not attempt to do, providing a candidate cognitive model besides the best performing model.

**Weaknesses:**

While the domains used here are interesting for a first study, they are still relatively simple. Most compelling would be to find a task with a richer space of hypotheses and experiments, perhaps also making better use of common sense priors embedded in LLMs.

The paper also doesn't provide much qualitative insight into what behavioral differences arise from doing online experiments compared to only doing batched inference, perhaps with refinement. While it might be hard to describe general differences, at least having a few example traces in the appendix (e.g., showing example scenarios where batched inference falls short, or outperforms the online model) would help us at least form some qualitative hypotheses on where the gains are coming from.

**Questions:**

* Are any of the 10 Zendo predicates particularly unlikely to be recovered? I'd be curious to see an example of the model recovering 'a red is bigger than all non reds', for instance, since this seems like a quite unlikely hypotheses to propose by itself.
* Is the hardcoded random experiment proposer biased in any useful way? It would be good to describe it at least in the appendix. Again, for most of the Zendo rules, it'd guess that most single random experiments would be highly non informative.
* In the result in Table 4, with a single proposal, there's nothing for the InfoGain reranking to do. So should the first column be interpreted as two random repeats of the same exact model?
* Appendix A1: missing an integral sign in (10)?

**Limitations:**

Yes, adequately addressed.

---

> ### Author Rebuttal · Authors · 2024-08-05
>
> We thank the reviewer for the positive feedback! Below, we address concerns and questions raised by the reviewer.
>
> > Concern 1: While the domains used here are interesting for a first study, they are still relatively simple.
>
> We fully acknowledge this concern.
>
> Our work evaluates the models on traditional, well-studied Bayesian concept learning datasets to demonstrate that using natural language representation is beneficial even in domains where crafting a DSL is still possible and also to enable in-depth analyses.
>
> We believe that our work is needed to convince the Bayesian concept learning community to consider using natural language representation, and it is also an important stepping stone towards fully wielding the power of natural language.
>
> We do show the almost obvious flexibility of natural language representation in our “majority of blocks are red” experiment. We leave evaluations on more complex domains where natural language representation is a must for future work.
>
> > Concern 2: The paper also doesn't provide much qualitative insight into what behavioral differences arise from doing online experiments compared to only doing batched inference
>
> Please see our global response where we provide actual hypothesis traces from our models. They will be included in our revised paper.
>
> > Question 1: Are any of the 10 Zendo predicates particularly unlikely to be recovered?  I'd be curious to see an example of the model recovering 'a red is bigger than all non reds', for instance
>
> There are two Zendo rules (out of ten) that are unlikely to be recovered by our models: ‘a red is bigger than all non reds’ and ‘all are blue or small’. We note that human participants from Bramley et al. [1] also perform very poorly on these two rules, as shown in [1]'s Figure 4.
>
> > Question 2: Is the hardcoded random experiment proposer biased in any useful way? It would be good to describe it at least in the appendix. Again, for most of the Zendo rules, it'd guess that most single random experiments would be highly noninformative.
>
> We did try to come up with a random experiment proposer that would produce useful experiments. Here is a description of it, which we will include in the appendix:
> Sample number of cones uniformly from 1-5
> Sample color, size, orientation, and groundedness uniformly
> Sample number of touchings from geometric-distribution(p=0.3) - 1
> Then, sample random pairs of blocks and make them touch until we have the specified number of touchings.
>
> > Question 3: In the result in Table 4, with a single proposal, there's nothing for the InfoGain reranking to do. So should the first column be interpreted as two random repeats of the same exact model?
>
> The rows in the first column still use different experiment proposers, so they are not random repeats.
>
>
> > Question 4: Appendix A1: missing an integral sign in (10)?
>
> Thank you for pointing it out! We will fix it.
>
> [1] Bramley et al. 2018, Grounding compositional hypothesis generation in specific instances.

---

> > ### Comment · Reviewer_i693 · 2024-08-10
> >
> > Thanks for the response. I appreciate the clarifications. I think commenting in the paper about what is hard about the two hardest Zendo rules would be useful to draw attention for future work.

---

### Author Rebuttal · Authors · 2024-08-05

We thank all the reviewers for their detailed feedback. We have posted responses to each reviewer's individual comments. Here, we address some common concerns raised by the reviewers.

> Common concern 1: Want to see qualitative differences, i.e., actual hypotheses proposed by the models

Below are example hypothesis traces of online, batch with refinement, and batch (without refinement) models on the Zendo task with the rule “a blue touch a red”. We will include them in our revised paper.

- Batch: 'Blocks must touch at least one other block' is proposed but is immediately falsified by an existing experiment where a scene with no blocks touching is negative.
- Batch with refinement: ‘Blue blocks must not touch green blocks’ is first proposed and then immediately refined into ‘Blue blocks must not touch blocks of any color other than red’ since there is an existing data point where a scene with a blue touching a blue is negative. This hypothesis later gets falsified, without an opportunity to refine itself since the model is not online, when a scene with no blocks touching turns out to be negative.
- Online: ‘There must be a blue block’ is proposed and added to the pool of particles. Since it has higher prior than other particles (has shorter length); it keeps surviving while others get killed, despite some conforming with the data. Upon seeing a scene with a blue touching a green being negative, the particle ‘There must be a blue block’ is perturbed into `there must be a blue block touching a red block’.

As you can see, the batch without refinement model lacks the ability to propose hypotheses that are consistent with existing data. The batch with refinement model fixes the issue but lets many of these almost-correct hypotheses get falsified without having a chance to refine itself upon seeing new experiments. The online models are the only models whose hypothesis proposal is affected by the prior. As defined by the prior, shorter hypotheses are preferred over longer hypotheses (the longer ones get killed), given that both explain data equally well. Once shorter hypotheses get falsified by a new experiment, they get perturbed into longer hypotheses that fit the data better.

While in principle the batch with refinement model could also do the ‘there must be a blue block’ -> ‘there must be a blue block touching a red block’ transition like the online model, the influence of prior in hypothesis proposal and the ability to revise existing hypotheses upon seeing new experiments is what differentiates online models from batch with refinement models.

> Common concern 2: Baselines are not described in detail. It is hard to understand the baseline without reading prior work.

We include more baseline details below and will include them in our revised paper.

In probabilistic inference terms, both batch with and without refinement correspond to importance sampling $p(h|x_{1:t},y_{1:t}) = E_{p(h’|x_{1:t},y_{1:t})} [1[h = h’]] =  E_{q(h’|x_{1:t},y_{1:t})} [\frac{p(h’|x_{1:t},y_{1:t})}{q(h’|x_{1:t},y_{1:t})}1[h = h’]]$.

The difference in the two baselines, batch without and with refinement, lies in how $q(h’|x_{1:t},y_{1:t})$ is constructed.

- Batch: $q(h|x_{1:t},y_{1:t}) = U(LLM(x_{1:t},y_{1:t}))$ where $LLM(...)$ prompts an LLM to return a list of hypotheses

- Batch with refinement: $q(h|x_{1:t},y_{1:t}) = U(Refined\text{-}LLM(x_{1:t},y_{1:t}, None, 0))$ where $Refined\text{-}LLM$ is defined as follows:

First, let $s(h, x_{1:t}, y_{1:t}) = \frac{1}{t}  \sum_{i=1}^t \mathbb{1}[h(x_i) = y_i]$. This simply scores what percentage of data points in $x_{1:t}, y_{1:t}$ that $h$ makes correct predictions.

function $Refined\text{-} LLM(x_{1:t}, y_{1:t}, h, k)$:

&nbsp;&nbsp;&nbsp;&nbsp; $H = LLM\text{-}with\text{-}h(x_{1:t},y_{1:t}, h)$ # Prompts LLM to refine h

&nbsp;&nbsp;&nbsp;&nbsp; if $k=K$:

&nbsp;&nbsp;&nbsp;&nbsp;&nbsp;&nbsp;&nbsp;&nbsp; return $\emptyset$

&nbsp;&nbsp;&nbsp;&nbsp; else if $\exists h’ \in H, s(h’, x_{1:t}, y_{1:t}) = 1$:

&nbsp;&nbsp;&nbsp;&nbsp;&nbsp;&nbsp;&nbsp;&nbsp; return {$h’ \in H | s(h’, x_{1:t}, y_{1:t}) = 1$}

&nbsp;&nbsp;&nbsp;&nbsp; else:

&nbsp;&nbsp;&nbsp;&nbsp;&nbsp;&nbsp;&nbsp;&nbsp; $h^* = argmax_{h’ \in H} (s(h’, x_{1:t}, y_{1:t}))$

&nbsp;&nbsp;&nbsp;&nbsp;&nbsp;&nbsp;&nbsp;&nbsp; return $Refined-LLM(x_{1:t}, y_{1:t}, h^*, k + 1)$

where $K$ is the number of refinements allowed.

> Common concern 3: The domains, while interesting and good for first study, are relatively simple

We fully acknowledge this concern.

Our work evaluates the models on traditional, well-studied Bayesian concept learning datasets to demonstrate that using natural language representation is beneficial even in domains where crafting a DSL is still possible and also to enable in-depth analyses.

We believe that our work is needed to convince the Bayesian concept learning community to consider using natural language representation, and it is also an important stepping stone towards fully wielding the power of natural language.

We do show the almost obvious flexibility of natural language representation in our “majority of blocks are red” experiment. We leave evaluations on more complex domains where natural language representation is a must for future work.

[1] Qiu et al. 2024, Phenomenal Yet Puzzling: Testing Inductive Reasoning Capabilities of Language Models with Hypothesis Refinement

[2] Wang et al. 2024, Hypothesis Search: Inductive Reasoning with Language Models.

---

### Decision · Program_Chairs · 2024-09-25

**Decision:**

Accept (poster)

**Comment:**

The paper introduces a novel approach to learn natural language hypotheses and conduct active experimentation for hypothesis refinement. It leverages LLMs for hypothesis generation and evaluation within the framework of Sequential Monte Carlo (SMC) and Bayesian Optimal Experimental Design (BOED). Experiments on two cognitive psychology tasks demonstrate improvements over existing methods and some agreement with human behavior.

The paper is strong because its method is technically principled and experiments are carefully carried out.

The main weakness is that the experimental settings are restricted, thus not fully utilizing the power of LLMs. Reviewers have raised questions in this aspect, and called for more analysis. Authors explain that experiments in simple domains are "important stepping stone towards fully wielding the power of natural language." The rebuttal also includes some new results and analysis. k

I think the strengths of this paper overweigh its weaknesses, and thus recommend to accept it. But I strongly recommend the authors to clarify their technical motivation and experiment design, as well as include new results and analysis in the camera-ready.